# Effects of high-intensity interval training on physical morphology, cardiorespiratory fitness and metabolic risk factors of cardiovascular disease in children and adolescents: A systematic review and meta-analysis

Jie Men[1‡]*, Shuangling Zou[1‡], Jia Ma[1], Chenmin Xiang[1], Shufeng Li[1], Junli Wang[2]

1 Department of Medical Laboratory Science, Fengyang College, Shanxi Medical University, Shanxi, China,
2 Xinjiang University, Xinjiang, China

‡ JM and SZ are co-first authors and contributed equally to this work.
* menjie2020@126.com

## Abstract

### Objective

To systematically evaluate the safety and efficacy of high-intensity interval training in children and adolescents.

### Methods

Eight databases were searched. Descriptive analysis of the efficacy and safety of high-intensity interval training on body shape, cardiorespiratory fitness, and metabolic risk markers of cardiovascular disease in children and adolescents. Subgroup analysis was performed using age, participants, intervention time, and exercise frequency as covariates.

### Results

47 studies included 2995 children and adolescents. The results of the meta-analysis showed that high-intensity interval training significantly improved cardiorespiratory fitness indicators ($VO_{2max}$, SBP, DBP and $HR_{max}$) and cardiovascular disease biomarkers (TC and HDL-C). HIIT had no significant effect on body shape indicators (BMI, BF% and WC) or cardiovascular disease biomarkers (TG and LDL-C).

### Conclusion

Currently, there is insufficient evidence that HIIT with interval running as the predominant form improves physical indicators in children and adolescents. However, HIIT can be promoted in children and adolescents to improve cardiorespiratory fitness and reduce some metabolic risk of cardiovascular disease.

**Data Availability Statement:** All relevant data are within the paper and its Supporting Information files.

**Funding:** Our manuscript was jointly funded by the Shanxi Provincial Department of Education's Higher Education Teaching Reform, Innovation Project (No. J2021967) and the Fenyang College Teaching Reform and Innovation Project of Shanxi Medical University (No. FJ202013) and and 2022 Research Project of Fenyang College of Shanxi Medical University (No.2022A01). FJ202013 and 2022A01 provide platform resources in terms of data collection and analysis. J2021967 provided constructive advice on publication decisions and manuscript preparation.

**Competing interests:** The authors have declared that no competing interests exist.

## Introduction

In 2018, the World Health Organization (WHO) conducted a summary analysis of 16 million children and adolescents in 146 countries and regions based on 298 population surveys and pointed out that 85% of girls and 78% of boys in the world did not meet the WHO recommended standards, which leads to an inevitable global trend towards insufficient physical activity in adolescents [1]. There is growing evidence that physical inactivity in children and adolescents increases the prevalence of cardiovascular disease in adulthood and affects cognitive development, social interaction and current and future health [2, 3]. Insufficient physical activity in children and adolescents is highly correlated with metabolic diseases in adulthood, especially with an increased risk of diseases such as metabolic obesity, type 2 diabetes mellitus ($T_2DM$), cardiovascular disease (CVD), and cancer [4]. To achieve the goal of reducing the insufficient rate of physical activity by 15% in 2030, it is necessary to enhance the level of physical activity of children and adolescents [1]. What we all know is that continuous aerobic exercise can increase the aerobic capacity of the body, improve the sensitivity of insulin resistance, improve the level of lipometabolism and reduce the risk of diseases caused by physical inactivity. However, aerobic exercise lasts for a long time and has a single rhythm, making it difficult for most people to persist. Nevertheless, one of the main obstacles to achieving regular physical activity for current children and adolescents is lack of time. Therefore, it is likely to replace aerobic exercise with high-intensity interval training (HIIT) because it has the advantages of low time cost, low exercise volume, easy persistence, and an equivalent exercise effect with aerobic exercise. HIIT refers to a training method that is repeated multiple times at greater than or equal to the anaerobic threshold or maximal lactate steady-state intensity with incomplete recovery between each set of exercises. The body is more sensitive to HIIT stimulation and produces more comprehensive benefits in terms of sports ability, skeletal muscle metabolism, and energy consumption.

In recent years, the comparison between the effects of HIIT and continuous aerobic exercise has become a hot research topic. The research groups are mainly obese people and athletes, who focus on body composition, metabolism, and cardiorespiratory fitness (CRF), whereas athletes are primarily concerned with athletic performance and physiological adaptation during exercise [6].

Research on HIIT in obese children and adolescents and normal children and adolescents has gradually attracted attention and achieved certain research results. A recently published meta-analysis of HIIT targeting obese children and adolescents showed that HIIT was effective in improving cardiometabolic level, cardiorespiratory adaptability, and aerobic capacity of obese children and adolescents, but the evidence for conclusions about body composition improvement is insufficient [2]. A meta-analysis of healthy children and adolescents suggested that HIIT is not only effective in improving the health of children and adolescents but also in improving cardiovascular disease risk factors [3]. A meta-analysis of young athletes showed that HIIT can improve aerobic and anaerobic exercise abilities of young athletes, and that the time cost is lower. Comparing HIIT with moderate-intensity continuous training (MICT) found similar effects on body composition, blood pressure in childhood obesity [4] and greater improvements in cardiorespiratory fitness in children and adolescents. HIIT can be used as an alternative training mode of MICT to maintain cardiometabolic health and can be applied to the management of childhood obesity. However, previous studies have small sample sizes, deviation of outcome index measurement tools, language bias [3], and unclear description of exercise dose [5], especially the lack of subgroup analyses on the influence of pre-puberty and pubertygender [2], which affect the stability of results. Given the above, this study will systematically evaluate the effectiveness and safety of HIIT for children and adolescents, expecting to provide a scientific basis for the promotion of HIIT in children and adolescents.

## Methods

### Protocol

A systematic review and meta-analysis were conducted in accordance with the 27 checklists applying the established guidelines of the PRISMA statement 2020 [6], whose aim is to serve as a basis for reporting systematic reviews of randomized trials. The search and method were pre-specified, and we have pre-registered with PROSPERO(10.37766/inplasy2022.10.0092).

### Document retrieval strategy

The computer retrieved PubMed, The Cochrane Library, Embase, Web of Science, Science Direct, CNKI, WanFang, and VIP databases. In addition, research published by Google Scholar was manually searched. Randomized and non-randomized controlled trials on the health efficacy and safety of HIIT between children and adolescents were collected. The retrieval time limit was from the establishment of the database to January 1, 2022. We employed the following MeSH terms: high-intensity interval training, high-intensity interval, high-intensity intermittent, adolescence, teenagers, randomized controlled trials, RCT, etc. Taking PubMed as an example, the specific search strategy is shown in Table 1.

### Inclusion and exclusion criteria

Studies on children and adolescents were considered for the systematic review, provided they met the following inclusion criteria:

- Type of study: Randomized controlled trial (RCT) or controlled trial.

- Participants: Children and adolescents aged 5–19 years (normal weight, obesity, disease, etc.).

- Interventions: The experimental group underwent high-intensity interval training, and the interventions had no specific requirements except for intensity (intensity $\geq$ 80% $HR_{max}$, $\geq$ 100% aerobic speed, or $\geq$ 80% $VO_{2max}$).

- Comparisons: The control group received no intervention.

**Table 1. Full-search strategy for PubMed.**

| Number | Search terms |
|---|---|
| #1 | Adolescent [MeSH Terms] |
| #2 | (((((Adolescence [Title/Abstract]) OR (Teens [Title/Abstract])) OR (Teenagers [Title/Abstract])) OR (Youths [Title/Abstract])) OR (Female Adolescents [Title/Abstract])) OR (Male Adolescents [Title/Abstract]) |
| #3 | #1 or #2 |
| #4 | High-Intensity Interval Training [MeSH Terms] |
| #5 | ((((High-Intensity Interval [Title/Abstract]) OR (High-Intensity Intermittent [Title/Abstract])) OR (High-Intensity Intermittent Exercises [Title/Abstract])) OR (Sprint Interval Trainings [Title/Abstract])) OR (HIIT[Title/Abstract]) |
| #6 | #4 or #5 |
| #7 | randomized controlled trial [MeSH Terms] |
| #8 | (((RCT[Title/Abstract]) OR (Randomized [Title/Abstract])) OR (Randomized CIinical [Title/Abstract])) OR (Controlled CIinical Trials [Title/Abstract]) |
| #9 | #7 or #8 |
| #10 | #9 AND #6 AND #3 |

- Outcomes: Body shape indicators, CRF indicators, and cardiovascular disease biomarkers.

  Studies were excluded for the following reasons:

- Not reported in Chinese or English.

- Controlled experiment before and after intervention.

- Duplicate published literature.

- Studies that could not extract important outcome data.

## Literature screening and data extraction

Two researchers (Shuangling Zou and Chenmin Xiang) independently screened the literature, extracted and cross-checked the data. If there was any disagreement, it would be resolved through consultation. If additional information was required, the corresponding author was contacted via email. The extracted content included: (i) basic information of the included studies: research title, author, publication year, journal name, etc. (ii) baseline characteristics and interventions of the study subjects. (iii) outcome index data and outcome index measurement methods. (iv) Whether lost to follow-up, withdrawal, medical supervision measures and description of adverse reaction events, etc.

## Risk of bias and evaluation of literature quality included in the study

The bias risk assessment tools Cochrane (RoB2.0) and RevMan 5.3 independently evaluated the risk of bias in the included studies by two investigators (Jie Men and Jia Ma) and cross-checked the results.

## Statistical analysis

Meta-analysis was performed using RevMan 5.3 and Stata 15.0. Mean difference (MD) was used as the effect analysis statistic, and each effect amount provided a 95% confidence interval (CI). Sensitivity analysis of the included studies was performed to assess data robustness, and the magnitude of heterogeneity was evaluated in combination with $I^2$: $I^2 < 25\%$ was low heterogeneity, $I^2 = 25–50\%$ was moderate heterogeneity, $I^2 > 50\%$ was high heterogeneity and the level of meta-analysis was set to $\alpha = 0.05$. If the heterogeneity between the results was not statistically significant, a fixed-effects model was used for meta-analysis; if there was statistical heterogeneity between the studies, a random-effects model was used for meta-analysis and subgroup analysis was used to further analyze the sources of heterogeneity. Publication bias was graphically assessed by Egger's linear regression analysis.

# Results

## Literature screening process and results

The literature search began in December 2021 and resulted in a total of 3,868 studies: 471 studies were deleted for duplication, and after title and abstract screening, 3195 studies were considered ineligible and 202 full texts were screened based on inclusion/exclusion criteria. Finally, 47 studies were included, and the content of the literature selection process and results are shown in Fig 1.

## Incorporate basic characteristics of research

Table 2 summarizes the basic characteristics of the 47 studies [7–53], which investigated 2995 subjects (HIIT group: 1749, control group: 1246). Among them, there were 1165 boys

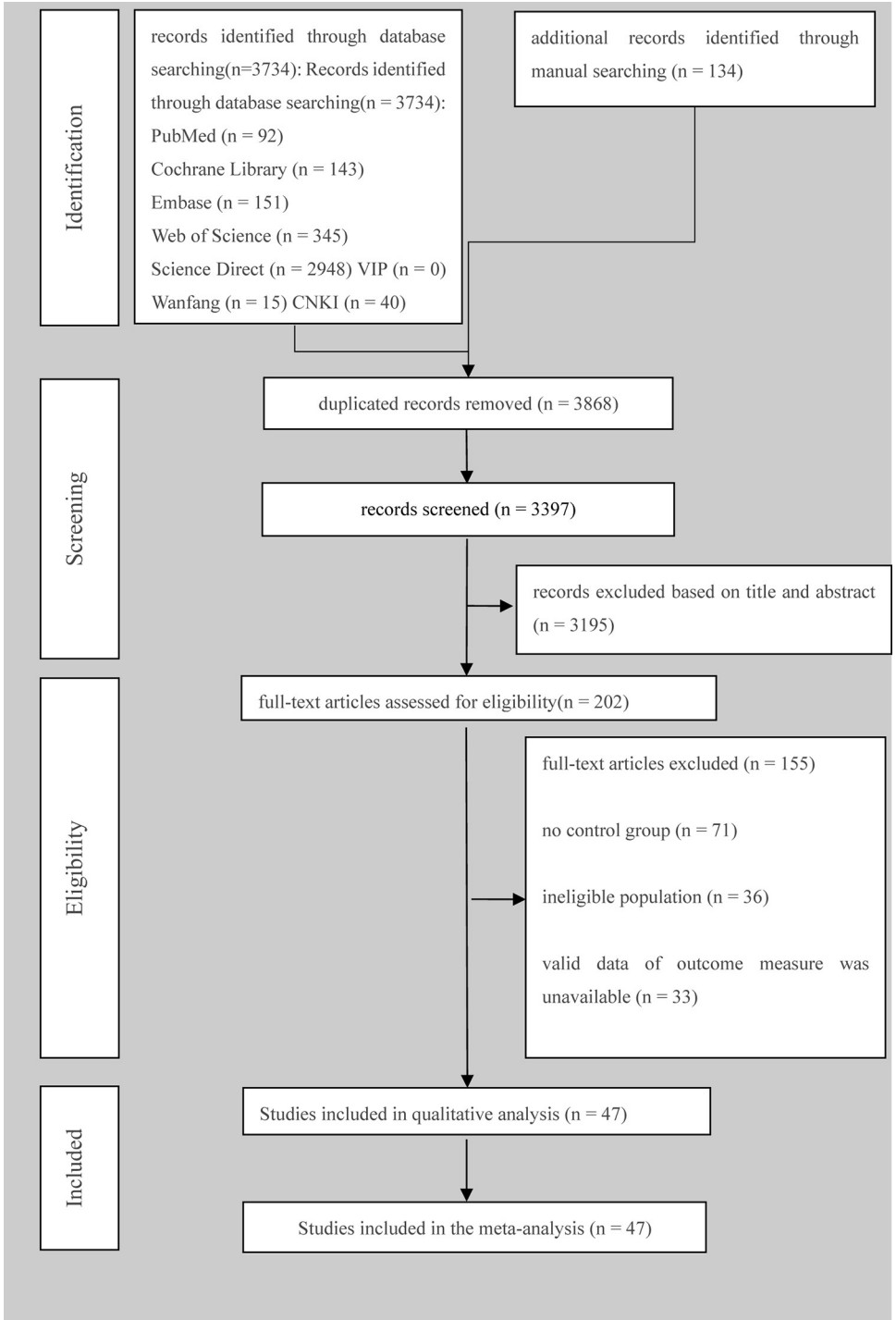

**Fig 1. PRISMA diagram outlining the results of the screening and selection.**

(38.90%), 1156 girls (38.60%) and 696 girls (23.24%) who did not mention sex. There were 2328 children (77.73%) and 667 adolescents (22.27%). 438 were overweight/obese (14.62%); 106 athletes (3.54%); 433 sick children and adolescents (14.46%); There were 26 studies with

**Table 2. Basic features of the included studies.**

| Author | Year | Country | Lesion | Age (experimental group) | Age (control group) | Sample (experimental group male/female) | Sample (control group male/female) | Intervention measure (experimental group) | Intervention measure (control group) | Intervention time (experimental group) | Intervention time (control group) | outcome indicator |
|---|---|---|---|---|---|---|---|---|---|---|---|---|
| Valérie et al. [7] | 2022 | France | obesity | $13.0 \pm 1.1$ | $13.2 \pm 1.0$ | 19 (11/8) | 11 (6/5) | mode of motion: ergometer bicycle exercise time:15mins exercise frequency: twice/week exercise intensity:75% to 90% $VO_{2\,max}$ | not any physical training | 16 weeks | 16 weeks | ①,② |
| Engel et al [17]. | 2019 | Germany | healthy | $11.6 \pm 0.2$ | $11.7 \pm 0.3$ | 17 (11/6) | 18 (11/7) | mode of motion: micro-session of Functional HIIT exercise time:>6 mins exercise frequency: week 1:3 times; week 2 to 4:4 times exercise intensity: >85% $HR_{max}$ | mode of motion: regular school class | 4 weeks | 4 weeks | ① |
| Georges et al. [46] | 2010 | France | healthy | $10.3 \pm 9.8$ | $10.1 \pm 1.2$ | 22 | 19 | mode of motion: interval run exercise time:18 to 39 minutes exercise frequency:3 additional PE and 2 regular mandatory PE exercise intensity:80 to 85% of MAS | mode of motion: regular mandatory PE exercise time:60 mins exercise frequency: 2/ week | 7 weeks | 7 weeks | ①,⑧,⑪ |
| MCNARRY et al. [10] | 2020 | England | asthma | Asthma: $14.1\pm0.9$ No-asthma: $14.1\pm0.8$ | Asthma: $14.2\pm1.0$ No-asthma: $13.9\pm0.9$ | Asthma group: 18 (10/8) No-asthma: 17 (9/8) | Asthma group: 18 (11/7) No-asthma: 16 (9/7) | mode of motion: games-based activities informed by formative work exercise time:30 mins exercise frequency:3/ week exercise intensity: >90%$HR_{max}$ | mode of motion: exercise time:30 mins | 6 mouths | 6 mouths | ①,⑧ |
| Mazurek et al. [37] | 2014 | Poland | females | $19.5\pm0.6$ $19.5\pm0.6$ | | 24 | 42 | mode of motion: Mechanically-braked cycle ergometers exercise time:47 mins exercise frequency:3/ week exercise intensity:60% $HR_{max}$ | mode of motion: regular PE exercise time:47 mins exercise frequency: 1/ week | 8 weeks | 8 weeks | ①,③,④,⑤,⑥,⑦,⑧ |
| Plavsic et al. [11] | 2020 | Serbia | obesity | $16.6 \pm 1.3$ | $15.8 \pm 1.5$ | 22 | 22 | mode of motion: Diet + HIIT (electronically braked cycle ergometer) exercise time:43mins exercise frequency:2/ week exercise intensity:85 to 90% of $HR_{max}$ | mode of motion: Diet | 12 weeks | 12 weeks | ①,②,③,④,⑤,⑥,⑦,⑧,⑨,⑩,⑪ |

(*Continued*)

**Table 2.** (Continued)

| Author | Year | Country | Lesion | Age experimental group | Age control group | Sample experimental group (male/female) | Sample control group (male/female) | Intervention measure experimental group | Intervention measure control group | Intervention time experimental group | Intervention time control group | outcome indicator |
|---|---|---|---|---|---|---|---|---|---|---|---|---|
| Cvetković et al. [24] | 2018 | Serbia. | obese males | | | 10 | 11 | mode of motion: PE +HIIT (interval runs) exercise frequency:2/week | mode of motion: PE exercise frequency: 2/week | 12weeks | 12weeks | ①,②,⑨,⑩,⑪ |
| Leahy et al. [18] | 2019 | Australia | healthy | 16.2±0.4 | 16.2±0.4 | 38 | 30 | mode of motion: included a combination of aerobic-based and resistance-based exercise time:12–20 min exercise frequency:3/week exercise intensity:>85% $HR_{max}$ | | 14 weeks | 14 weeks | ①,⑧ |
| Costigan et al. [29] | 2016 | UK | healthy | 15.7±0.7 | 15.6±0.6 | 21 | 22 (11/11) | mode of motion: gross motor cardiorespiratory exercises exercise time:8-10mins exercise frequency:3/week | mode of motion: PE exercise frequency:3/week | 8 weeks | 8 weeks | ① |
| Rosenkranz et al. [43] | 2012 | Manhattan | healthy | 8.8±0.6 | 9.8±4.1 | 8 | 8 | mode of motion: performed on an indoor track exercise time:30mins exercise frequency: sessions spaced at least 48 h exercise intensity:100–130% of MAS | | 8 weeks | 8 weeks | ①,③,④,⑤,⑥,⑦,⑨,⑩ |
| Hammami et al. [25] | 2018 | Tunisia | soccer players | 15.9 ± 0.4 | 15.8 ± 0.7 | 10 | 10 | mode of motion: small-sided soccer drills exercise time:35-45mins exercise frequency:2/week exercise intensity: ~ 85% $HR_{max}$ | mode of motion: regular physical activity exercise frequency: 2/week | 8 weeks | 8 weeks | ①,② |
| Winn et al. [8] | 2021 | UK | asthma | Asthma: 13.7 ± 1.0 No-asthma: 13.8±1.1 | Asthma: 13.4±1.2 No-asthma: 13.5±1.0 | 221 (116/105) | 69 (21/48) | mode of motion: game-based activities exercise frequency:3/week exercise time:30mins exercise intensity: >90% $HR_{max}$ | mode of motion: incremental ramp test | 6 mouths | 6 mouths | ① |

*(Continued)*

**Table 2.** (Continued)

| Author | Year | Country | Lesion | Age | | Sample | | Intervention measure | | Intervention time | | outcome indicator |
|---|---|---|---|---|---|---|---|---|---|---|---|---|
| | | | | experimental group | control group | experimental group (male/female) | control group (male/female) | experimental group | control group | experimental group | control group | |
| Malte Nejst Larsen et al. [12] | 2020 | Denmark | healthy | 10±0.3 | 10±0.3 | 57 | 115 | exercise time:12 mins exercise frequency:5/week | mode of motion: PE lessons | 10 mouths | 10 mouths | ⑨,⑩ |
| Soori et al. [13] | 2020 | Iran | hyperactivity | 12.55 ± 0.15 | 12.5 ± 0.45 | 26 (9/17) | 17 (11/6) | mode of motion:20 meters running program exercise time:>10 mins exercise frequency:3/week exercise intensity: 85% $HR_{max}$ | | 6 weeks | 6 weeks | ⑪ |
| Racil et al. [40] | 2013 | Tunisian | obese | 15.6 ± 0.7 | 15.9 ± 1.2 | 11 | 12 | mode of motion: short bursts exercise frequency:3/week exercise intensity:100 to 110% of MAS | non-exercising | 12weeks | 12 weeks | ②,③,④,⑤,⑥,⑦,⑧ |
| Lambrick et al. [30] | 2016 | UK | healthy and obese | Obesity: 9.3±0.8 Normal: 9.2±0.7 | Obesity: 9.4±0.8 Normal: 9.2±0.8 | Normal:13 (8/5) Obesity:15 (10/5) | Normal: 13 (7/6) Obesity: 14 (7/7) | mode of motion: equipment exercise time:40 min exercise frequency:2/week exercise intensity: 40% difference between GET and $VO_{2max}$ | mode of motion: PE | 8 weeks | 8 weeks | ①,②,③,⑧, ⑪ |
| Baquet et al. [52] | 2001 | France | health | 12.8 ± 1.2 | 13.5 ± 0.9 | 503 (263/240) | 48 (21/27) | mode of motion: PE + running exercises. exercise time:3 h exercise frequency:3/week exercise intensity:100 to 120% MAS | mode of motion: PE exercise time:3 h frequency:3/week | 10 weeks | 10 weeks | ①,② |
| Tjønna et al. [49] | 2009 | Norway | obese | 14.0 ±0.3 | | 13 | 14 | mode of motion: walking/running exercise time:.40min exercise frequency: 2/week exercise intensity:90% $HR_{max}$ | mode of motion: exercise exercise frequency: 2/month | 3 months | 12 mouths | ①,②,③,⑧,⑨,⑩ |
| Zhu Kunru [14] | 2020 | China | healthy girl | 16.35±0.490 | 17.20±0.410 | 20 (0/20) | 20 (0/20) | mode of motion: rope skipping exercise time:15-20mins exercise frequency:3/week | Regular training 15-20mins | 12 weeks | 12 weeks | ① |

(Continued)

**Table 2.** (Continued)

| Author | Year | Country | Lesion | Age | | Sample | | Intervention measure | | Intervention time | | outcome indicator |
|---|---|---|---|---|---|---|---|---|---|---|---|---|
| | | | | experimental group | control group | experimental group (male/female) | control group (male/female) | experimental group | control group | experimental group | control group | |
| Li kang [26] | 2018 | China | healthy girl | 16.34±0.91 | 16.63±0.90 | 38 (0/38) | 54 (0/54) | mode of motion: sprints exercise time:4-9mins (Stepwise increase) exercise frequency:2/week | Regular PE | 8 weeks | 8 weeks | ①,⑧ |
| Mu Taiyang [19] | 2019 | China | overweight and obese male | 17.00±0.89 | 17.18±0.98 | 11 (11/0) | 11 (11/0) | mode of motion: Running and judo training exercise time:45mins exercise frequency:3/week exercise intensity:>85% $HR_{max}$ | Regular training 45mins | 12 weeks | 12 weeks | ①,②,③ |
| Yang Zhongwu [20] | 2019 | China | teenagers | 11.50±0.513 | 11.50±0.513 | 20 (10/10) | 20 (10/10) | exercise time:45mins exercise frequency:2/week exercise intensity:85% to 90% $HR_{max}$ | 60% to 70% $HR_{max}$ | 8 weeks | 8 weeks | ①,③ |
| Huo Kaiwen [15] | 2020 | China | teenagers | boy:12.70 ±0.48 girl:12.50 ±0.53 | boy:12.80 ±0.42 girl:12.90 ±0.32 | 20 | 20 | exercise time:50-60mins exercise frequency:2/week exercise intensity:90% to 95% $HR_{max}$ | Regular training 50 to 60mins | 8 weeks | 8 weeks | ①,⑧ |
| Ma Qin [16] | 2020 | China | obese male adolescents | 13.53±0.72 | 13.90±0.89 | 15 (15/0) | 15 (15/0) | mode of motion: Combination of training exercise time:40-50mins exercise frequency:2/week exercise intensity:85% $HR_{max}$ | Regular PE | 8 weeks | 8 weeks | ①,② |
| Dai Xiangdi [9] | 2021 | China | healthy teenagers | About 14 | About 14 | 49 | 47 | mode of motion: Combination of training exercise time:20mins exercise frequency:2/week exercise intensity:65% to 85% $HR_{max}$ | Regular PE | 8 weeks | 8 weeks | ① |
| Cao et al. [44] | 2012 | China | obese adolescent boys | 13–15 | 13–15 | 20 (20/0) | 20 (20/0) | mode of motion: Combination of training exercise time:50-60mins exercise frequency:2/week exercise intensity:90% to 95% $HR_{max}$ | Daily habits | 8 weeks | 8 weeks | ①,②,⑧,⑨,⑩ |

(Continued)

**Table 2.** (Continued)

| Author | Year | Country | Lesion | Age | | Sample | | Intervention measure | | Intervention time | | outcome indicator |
|---|---|---|---|---|---|---|---|---|---|---|---|---|
| | | | | experimental group | control group | experimental group (male/ female) | control group (male/ female) | experimental group | control group | experimental group | control group | |
| Martin Smith et al. [21] | 2019 | Scottish | from 2 higher PE class | 17±0.3 | 16.8±0.5 | 22 (13/9) | 30 (19/11) | mode of motion: running sprints exercise time:25-26mins exercise frequency:3/week exercise intensity:92% of HR$_{max}$ | mode of motion: PE exercise time:1 h | 4 weeks | 4weeks | ①,③,④,⑤,⑥,⑦,⑧,⑨,⑩ |
| Ludyga et al. [22] | 2019 | Switzerland | healthy male adolescents | 14±0.8 | 13.9±0.6 | 32 | 28 | mode of motion: a circuit training exercise time:20 mins | | | | ①,⑪ |
| Ruiz–Ariza et al. [23] | 2019 | Spain | healthy | 13.79 ± 1.38 | 13.67 ± 1.29 | 90 (46/44) | 94 (52/42) | mode of motion: Combination of training exercise time:16mins exercise frequency:2/week exercise intensity: over 85% HR$_{max}$ | mode of motion: PE exercise frequency: 2/week | 12 weeks | 12 weeks | ① |
| Racil et al. [31] | 2016 | Tunis | obese adolescent females | 14.2±1.2 | | 17 | 14 | mode of motion: Combined interval running exercise time:>35mins exercise frequency:3/week exercise intensity:100% MAS | mode of motion: non-exercising group | 12 weeks | 12 weeks | ②,③,⑨,⑩,⑪ |
| Weston et al. [32] | 2016 | United Kingdom | healthy and obese | 14.1 ± 0.3 | 14.1 ± 0.3 | 41 (33/8) | 60 (30/30) | mode of motion: Many kinds of sports exercise frequency:3/week exercise intensity:90% HR$_{max}$ | mode of motion: PE exercise frequency: 3/week | 10 weeks | 10 weeks | ①,②,③,⑨,⑩ |
| Racil et al. [33] | 2016 | Tunisia | obese female adolescents | 16.6 ± 0.9 | 16.9 ± 1.0 | 23 | 19 | mode of motion: plyometric exercises exercise time: 26 to28mins exercise frequency:3/week exercise intensity:100%VO$_2$ $_{max}$ | no exercise | 12 weeks | 12 weeks | ②,③,⑧ |
| McNarry et al. [34] | 2015 | UK | health and obesity | Obesity: 9.3 ± 0.9 Normal: 9.2 ± 0.8 | Obesity: 9.3 ± 0.9 Normal: 9.2 ± 0.8 | Normal:13 Obesity:15 | Normal:16 Obesity:11 | mode of motion: physical activity exercise time:10 mins exercise frequency:2/week | usual care control group | 6 weeks | 6 weeks | ①,⑧,⑪ |

*(Continued)*

**Table 2.** (Continued)

| Author | Year | Country | Lesion | Age experimental group | Age control group | Sample experimental group (male/female) | Sample control group (male/female) | Intervention measure experimental group | Intervention measure control group | Intervention time experimental group | Intervention time control group | outcome indicator |
|---|---|---|---|---|---|---|---|---|---|---|---|---|
| Martin et al. [35] | 2015 | UK | healthy | $16.9 \pm 0.3$ | $16.8 \pm 0.6$ | 20 (13/7) | 23 (18/5) | mode of motion: Sprint combination training exercise time:60 mins exercise frequency:3/ week | mode of motion: PE exercise frequency: 3/ week | 7 weeks | 7 weeks | ① |
| Peter Riis Hansen et al. [41] | 2013 | Porto district, Portugal | overweight children | 8–12 | 8–12 | 20 (17/3) | 11 (7/4) | mode of motion: technical football exercises and small-sided football games exercise time:1h-1.5h exercise frequency:4/ week exercise intensity: > 80% $HR_{max}$ | mode of motion: compulsory sport curriculum at school exercise frequency: 2/week exercise time:45 to 90min/time | 3mouths | 3mouths | ①,⑨,⑩,⑪ |
| Patrick Mucci et al. [42] | 2013 | Tanner | prepubescent children | $10.3\pm0.7$ | $9.8\pm0.6$ | 9 (4/5) | 9 (6/3) | mode of motion: running exercise time:30mins exercise frequency:2/ week exercise intensity:100%-130% | usual physical activities | 8 weeks | 8 weeks | ⑧ |
| Buchan et al. [45] | 2012 | West of Scotland | youth adolescent | $16.7\pm0.1$ | $16.3\pm0.5$ | 17 (15/2) | 24 (20/4) | mode of motion: repetitions of maximal sprint running exercise time:54mins exercise frequency:3/ week exercise intensity: maximal sprint running | maintain normal activity patterns | 7weeks | 7weeks | ①,②,④,⑤,⑥,⑦,⑨,⑩ |
| Breil et al. [47] | 2010 | Bern, Switzerland | healthy elite junior alpine skiers | $17.4\pm1.1$ | $16.6\pm1.1$ | 13 | 8 | mode of motion: cycle ergometer; ski-speciWc obstacle running course containing slalom, balancing and jumping elements exercise frequency:15 times training exercise time:16mins exercise intensity:90–95% $HR_{max}$ | continued their normal endurance and strength training | 11days | 11days | ①,②,⑧,⑩ |

(*Continued*)

**Table 2.** (Continued)

| Author | Year | Country | Lesion | Age | | Sample | | Intervention measure | | Intervention time | | outcome indicator |
|---|---|---|---|---|---|---|---|---|---|---|---|---|
| | | | | experimental group | control group | experimental group (male/female) | control group (male/female) | experimental group | control group | experimental group | control group | |
| Ferrete et al. [39] | 2014 | Spain | healthy young soccer players | 9.32±0.25 | 8.26±0.33 | 11 | 13 | mode of motion: underwent soccer training; 1/4 squat, deep jumps, CMJ with weight, and sprint exercises exercise intensity: maximal voluntary intensity using player's body weight (or body weight plus light resistances) as external resistance exercise time:30mins exercise frequency:3/week | mode of motion: underwent soccer training exercise time:30mins exercise frequency:3/week | 26weeks | 26weeks | ② |
| Lau et al. [36] | 2015 | Hong Kong | overweight children | 11.0 ± 0.6 | 10.6 ± 0.6 | 15 | 12 | mode of motion: intermittent running; attended normal PE exercise time:72mins exercise frequency:3/week exercise intensity:120% of MAS | mode of motion: attended normal PE exercise time:35mins exercise frequency:2/week | 6weeks | 6weeks | ①, ⑪ |
| Boer et al. [38] | 2014 | Belgian | adolescents and young adults with intellectual disability | 18±3.2 | 17.4±2.4 | 17 (11/6) | 14 (9/5) | mode of motion: cycling exercise time:40mins exercise frequency:2/week exercise intensity: >100%VTR | participated in usual everyday scholar activities without supervised exercise training | 15 weeks | 15weeks | ①,②,③,④,⑤,⑨,⑩ |
| A. M. McManus et al. [50] | 2005 | Hong Kong | boys | 10.35±0.32 | 10.51 ± 0.3 | 10 | 15 | mode of motion: Loop pedalling exercise time:20mins exercise frequency:3time/week | normal physical activity | 8weeks | 8weeks | ⑧,⑪ |
| Helgerud et al. [53] | 2001 | America | male | 18.1±0.8 | 18.1±0.8 | 9 | 10 | mode of motion: run + game exercise frequency:2/week exercise intensity: 90 to 95% of HR$_{max}$ | Regular exercise | 8weeks | 8weeks | ⑧ |
| G. Baquet et al. [51] | 2002 | France | pubescent children | 9.7±0.9 | 10.1±0.4 | 20 (10/10) | 33 (13/20) | mode of motion: high intensity intermittent running exercises exercise time:30mins exercise frequency:2/week exercise intensity:100 to130% of MAS | Normal PE | 7weeks | 7weeks | ②,⑧,⑪ |

(Continued)

**Table 2.** (Continued)

| Author | Year | Country | Lesion | Age | | Sample | | Intervention measure | | Intervention time | | outcome indicator |
|---|---|---|---|---|---|---|---|---|---|---|---|---|
| | | | | experimental group | control group | experimental group (male/female) | control group (male/female) | experimental group | control group | experimental group | control group | |
| Anneke van Biljon et al. [27] | 2018 | The Republic | children | 11.1 ± 0.8 | 11.1 ± 0.8 | 29 | 24 | mode of motion: Sprint cycle exercise time:23mins exercise frequency:3/week exercise intensity: > 80% $HR_{max}$ | Normal PE | 5weeks | 5weeks | ①,③,⑧,⑨,⑩ |
| Juliana Pizzi et al. [28] | 2017 | America | obese adolescents | 12.18 ± 1.5 | 14.29 ± 1.8 | 20 | 34 | mode of motion: running exercise time:45mins exercise frequency:3/ two days | | 12weeks | 12weeks | ①,③,④,⑤,⑥,⑦ |
| Lynne Mary Boddy et al. [48] | 2010 | UK | obese | 11.78 ± 0.2 | 11.87 ± 0.3 | 8 | 8 | mode of motion: high intensity exercise program loosely based on dance exercise time:20mins exercise frequency:4/ week | Normal life | 3weeks | 3weeks | ①,②,③,⑧,⑨,⑩ |

Notes: ①BMI,②BF%,③WC,④TC,⑤TG,⑥HDL-C,⑦LDL-C,⑧$VO_2$,⑨SBP,⑩DBP, ⑪$HR_m$

medical supervision description (55.32%), 0 studies without medical supervision and 21 studies without detailed description (44.68%).

## Study quality assessment (risk of bias) and sensitivity analysis

A total of 47 studies with reasonable overall risk bias and high-quality papers were included in this meta-analysis (Fig 2). Egger's examined BMI, BF%, WC, $VO_{2max}$, SBP, DBP, and $HR_{max}$ and found that BF% was at a risk of publication bias (p<0.05) (Table 3). The sensitivity results showed that the overall data were stable (S1–S11 Figs).

## Subgroup analysis

Substantial sources of heterogeneity were explored using subgroup analysis. Due to differences in age, participants, intervention time, and exercise frequency of the HIIT intervention in children and adolescents included in the study, the HIIT assessment of body morphology, CRF and cardiovascular disease biomarkers may be affected. Therefore, subgroup analyses were performed based on age (5–15 years old, $\geq$ 15 years old), participants (health, obesity, else whose participants were not specified, not limited to healthy, obese people, etc.), intervention time ($\leq$ 10 weeks, > 10 weeks), and exercise frequency ($\geq$ 3 times/week, < 3 times/week), as shown in Table 4.

## Meta-analysis results

**Indicators of body morphology.** In 42 studies, HIIT (n = 1638) did not improve body morphology compared with the control group (n = 1080).

In 37 studies, the HIIT group (n = 1518) had no significant effect on BMI (MD = -0.30, 95% CI [-0.72, 0.13], p = 0.17) compared with the control group (n = 954). In 20 studies, the HIIT group (n = 829) had no significant effect on BF% (MD = -0.79, 95% CI [-1.64, 0.06], p = 0.07) compared with the control group (n = 394). In 16 studies, the HIIT group (n = 314) had no significant effect on WC (MD = -1.24, 95% CI [-2.78, 0.30]) compared with the control group (n = 359) (Fig 3).

## CRF indicators

In 32 studies, HIIT (n = 702) effectively improved CRF indices compared with the control groups (n = 791), but clinical heterogeneity was high; therefore, a subgroup analysis of CRF indices was conducted (Fig 4).

In 19 studies, the HIIT group (n = 379) effectively increased $VO_{2max}$ (MD = 2.91, 95% CI [1.80, 4.02], p < 0.001) compared with the control group (n = 416), but with higher heterogeneity ($I^2$ = 77%, p < 0.001). The results of the subgroup analysis showed that HIIT was more effective in aged 5–15 years (HIIT: n = 179, control group: n = 189), healthy children and adolescents (HIIT: n = 133, control group: n = 141), intervention time $\leq$ 10 weeks (HIIT: n = 244, control group: n = 292) and exercise frequency < 3 times/week (HIIT: n = 179, control group: n = 209)

In 14 studies, the HIIT group (n = 301) had effectively reduced SBP (MD = -2.73, 95% CI [-4.67, -0.79], p = 0.006) compared with the control group (n = 375), but with higher heterogeneity ($I^2$ = 65%, p < 0.001). The results of the subgroup analysis showed that HIIT was more effective in aged 5–15 years (HIIT: n = 223, control group: n = 285), else children and adolescents (HIIT: n = 104, control group: n = 122), intervention time $\leq$ 10 weeks (HIIT: n = 145, control group: n = 174) and exercise frequency $\geq$ 3times/week (HIIT: n = 219, control group: n = 294).

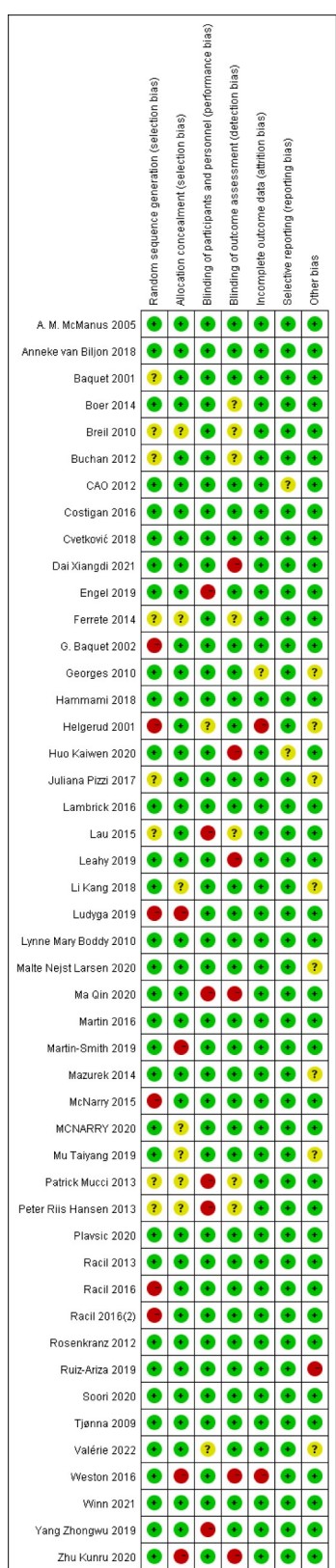

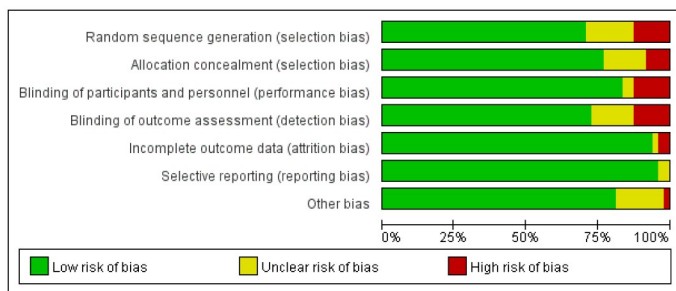

**Fig 2. Analysis of the risk of bias according to the Cochrane collaboration guideline.**

**Table 3. Three-line table of Egger's publishing bias.**

| Outcomes | Number of studies | Root MSE | Std. Eff. | Coef. | Std. Err. | t | p>\|t\| | [95% Conf. Interval] |
|---|---|---|---|---|---|---|---|---|
| BMI | 36 | 1.524 | slope | .2195473 | .2006643 | 1.09 | 0.282 | -.1882517 .6273462 |
| | | | bias | -1.013953 | .7345551 | -1.38 | 0.176 | -2.506749 .4788423 |
| BF% | 20 | 1.605 | slope | -.6003318 | .3335849 | 1.80 | 0.089 | -.100504 1.301168 |
| | | | bias | -2.485502 | 1.056721 | -2.35 | 0.030 | -4.70559 -.2654136 |
| WC | 16 | 1.673 | slope | .7849622 | .5614075 | 1.40 | 0.184 | -.4191371 1.9890611 |
| | | | bias | -3.023636 | 1.763666 | -1.71 | 0.108 | -6.806323 .7590503 |
| VO$_{2max}$ | 19 | 1.644 | slope | -.3187151 | .4612036 | -0.69 | 0.499 | -1.29177 .6543394 |
| | | | bias | 2.870097 | 1.416205 | 2.03 | 0.059 | -.1178346 5.858029 |
| SBP | 14 | 1.446 | slope | .078006 | .3490282 | 0.22 | 0.827 | -.6824612 .8384732 |
| | | | bias | -1.319617 | 1.166486 | -1.13 | 0.280 | -3.861171 1.221937 |
| DBP | 14 | 1.8 | slope | .0417849 | .4332558 | 0.10 | 0.925 | -.9021983 .9857682 |
| | | | bias | -.3722117 | 1.448374 | -0.26 | 0.802 | -3.527947 2.783524 |
| HR$_{max}$ | 13 | 2.569 | slope | -1.718386 | 1.203516 | -1.43 | 0.181 | -4.367307 .9305347 |
| | | | bias | 5.734588 | 3.564588 | 1.61 | 0.136 | -2.111017 13.58019 |

In 14 studies, the HIIT group (n = 301) effectively reduced DBP (MD = -2.42, 95% CI [-4.45, -0.38], p = 0.02) compared with the control group (n = 375), but with higher heterogeneity ($I^2$ = 70%, p < 0.001). The results of the subgroup analysis showed that HIIT was more effective in aged 5–15 years (HIIT: n = 223, control group: n = 285), else children and adolescents (HIIT: n = 104, control group: n = 122), intervention time ≤ 10 weeks (HIIT: n = 145, control group: n = 174) and exercise frequency ≥ 3 times/week (HIIT: n = 219, control group: n = 294).

In 12 studies, the HIIT group (n = 243) had an effective increase in HR$_{max}$ (MD = 5.91, 95% CI [1.24, 10.58], p = 0.01) compared with the control group (n = 233), but with higher heterogeneity ($I^2$ = 97%, p < 0.001). The results of the subgroup analysis showed that HIIT was more effective in children aged 5–15 years (HIIT: n = 208, control group: n = 101), healthy children and adolescents (HIIT: n = 95, control group: n = 82), intervention time > 10 weeks (HIIT: n = 49, control group: n = 47) and exercise frequency < 3 times/week (HIIT: n = 108, control group: n = 120).

### Cardiovascular disease biomarkers

In 8 studies, compared with the control group (n = 141), the HIIT group (n = 186) significantly improved metabolic risk index of cardiovascular disease, but some of the indicators were not statistically significant (Fig 5).

In 8 studies, HIIT (n = 141) effectively reduced TC compared to the control group (n = 186) (MD = -0.27, 95% CI [-0.38, -0.17], p < 0.001) with no significant heterogeneity ($I^2$ = 14%, p = 0.32). In 7 studies, compared with the control group (n = 172), HIIT (n = 124) effectively increased HDL-C levels (MD = 0.07, 95% CI [0.02, 0.12], p = 0.003) without significant heterogeneity ($I^2$ = 0%, p = 0.93). In 8 studies, the effect of HIIT on TG was not statistically significant between the HIIT (n = 141) and control group (n = 186) (MD = -0.00, 95% CI [-0.15, 0.14], p = 0.95). In 7 studies, the effect of HIIT on LDL-C was not statistically significant between the HIIT (n = 124) and control group (n = 172) (MD = -0.17, 95% CI [-0.34, 0.00], p = 0.05).

### Discussion

An increasing number of studies have revealed that physical activity can significantly improve the physical health of both children and adolescents. Nevertheless, global survey data show

**Table 4. Three-line table of subgroup analysis.**

| Outcomes | Subgroup | | The Number of studies | Pooled estimate [SMD/MD (95% CI)] | p value | $I^2$(%) | Test for subgroup differences |
|---|---|---|---|---|---|---|---|
| BF(%) | Participants | Health | 4 | 1.23 (-0.96, 3.42) | p = 0.270 | 42.0% | p = 0.020 |
| | | Obesity | 11 | -1.59 (-2.59, -0.58) | p = 0.002 | 77.0% | |
| | | Else | 5 | 0.12 (-1.11, 1.35) | p = 0.850 | 0.0% | |
| WC(cm) | Participants | Health | 2 | 0.57 (-3.54, 4.67) | p = 0.790 | 0.0% | p = 0.350 |
| | | Obesity | 8 | -2.06 (-3.26, -0.86) | p < 0.001 | 0.0% | |
| | | Else | 6 | -0.36 (-3.82, 3.09) | p = 0.840 | 83.0% | |
| VO$_{2max}$ | Age | 5 < < 15 | 10 | 3.99 (2.76, 5.22) | p < 0.001 | 44.0% | p = 0.130 |
| | | ≥ 15 | 9 | 1.76 (0.75, 2.76) | p = 0.090 | 91.0% | |
| | Participants | Health | 5 | 3.49 (1.57, 5.41) | p < 0.001 | 42.0% | p < 0.001 |
| | | Obesity | 6 | 2.12 (0.05, 4.18) | p = 0.040 | 92.0% | |
| | | Else | 8 | 3.31 (1.93, 4.69) | p < 0.001 | 0.0% | |
| | Intervention Time | ≤ 10 weeks | 14 | 3.59 (2.38, 4.81) | p < 0.001 | 44.0% | p = 0.130 |
| | | > 10 weeks | 5 | 1.77 (-0.26, 3.79) | p = 0.090 | 91.0% | |
| | Exercise Frequency | < 3 times/ week | 9 | 3.09 (1.80, 4.38) | p < 0.001 | 65.0% | p = 0.750 |
| | | ≥ 3 times/ week | 10 | 2.76 (1.21, 4.31) | p < 0.001 | 67.0% | |
| SBP | Age | 5 < < 15 | 10 | -2.00 (-4.22, 0.21) | p = 0.008 | 69.0% | p = 0.100 |
| | | ≥ 15 | 4 | -4.99 (-7.83, -2.15) | p < 0.001 | 0.0% | |
| | Participants | Health | 3 | -2.84 (-5.16, -0.52) | p = 0.020 | 23.0% | p = 0.910 |
| | | Obesity | 7 | -3.20 (-6.47, 0.07) | p = 0.006 | 73.0% | |
| | | Else | 4 | -1.89 (-6.70, 2.92) | p = 0.440 | 69.0% | |
| | Intervention Time | ≤ 10 weeks | 7 | -2.11 (-5.01, 0.79) | p = 0.150 | 60.0% | p = 0.530 |
| | | > 10 weeks | 7 | -3.43 (-6.30, 0.56) | p = 0.020 | 72.0% | |
| | Exercise Frequency | < 3 times/ week | 5 | -3.48 (-7.42, 0.46) | p = 0.080 | 75.0% | p = 0.067 |
| | | ≥ 3 times/ week | 9 | -2.48 (-4.78, -0.19) | p = 0.030 | 55.0% | |
| DBP | Age | 5 < < 15 | 10 | -1.59 (-3.93, 0.74) | p = 0.180 | 74.0% | p = 0.070 |
| | | ≥ 15 | 4 | -4.99 (-7.83, -2.15) | p < 0.001 | 0.0% | |
| | Participants | Health | 3 | -2.84 (-5.16, 0.52) | p = 0.020 | 23.0% | p = 0.940 |
| | | Obesity | 7 | -2.52 (-5.92, 0.89) | p = 0.150 | 79.0% | |
| | | Else | 4 | -1.89 (-6.70, 2.92) | p = 0.440 | 69.0% | |
| | Intervention Time | ≤ 10 weeks | 7 | -1.44 (-4.77, 1.90) | p = 0.400 | 73.0% | p = 0.37 |
| | | > 10 weeks | 7 | -3.43 (-6.3, -0.56) | p = 0.020 | 72.0% | |

(*Continued*)

**Table 4.** (Continued)

| Outcomes | Subgroup | | The Number of studies | Pooled estimate [SMD/MD (95% CI)] | p value | I²(%) | Test for subgroup differences |
|---|---|---|---|---|---|---|---|
| | Exercise Frequency | < 3 times/ week | 5 | -3.48 (-7.42, 0.46) | p = 0.080 | 75.0% | p = 0.54 |
| | | ≥ 3 times/ week | 9 | -2.00 (-4.63, 0.44) | p = 0.140 | 69.0% | |
| HR_max | Age | 5 < < 15 | 10 | 7.27 (2.12, 12.41) | p = 0.006 | 97.0% | p = 0.008 |
| | | ≥ 15 | 2 | -2.82 (-8.21, 2.57) | p = 0.300 | 0.0% | |
| | Participants | Health | 2 | 45.39 (-39.87, 130.65) | p = 0.300 | 100.0% | p = 0.570 |
| | | Obesity | 4 | -0.01 (-2.47, 2.46) | p = 1.000 | 41.0% | |
| | | Else | 6 | 0.22 (-1.50, 1.06) | p = 0.740 | 13.0% | |
| | Intervention Time | ≤ 10 weeks | 8 | 0.09 (-1.16, 1.34) | p = 0.890 | 14.0% | p = 0.650 |
| | | > 10 weeks | 3 | -0.48 (-2.67, 1.70) | p = 0.660 | 35.0% | |
| | Exercise Frequency | < 3 times/ week | 5 | -1.85 (-3.86, 0.16) | p = 0.070 | 0.0% | p = 0.030 |
| | | ≥ 3 times/ week | 6 | 0.45 (-0.02, 0.92) | p = 0.060 | 3.0% | |

that most children and adolescents do not meet the standards of physical activity guidelines, which seriously affects their current and future health [54]. To improve the level of physical activity of children and adolescents and achieve the goal of reducing the incidence of physical inactivity in children and adolescents by 15% in 2030 [1], WHO issued the updated "2020 WHO Guidelines on Physical Activity and Sedentary Behavior" in 2020, which recommended that children and adolescents should engage in an average of not less than 60 minutes of moderate-to vigorous-intensity physical activity per day. Because children and adolescents are concentrated on campus most of the time, considering that physical activity is limited by time, HIIT is a good choice because of its sports characteristics [55].

This study is the first to systematically evaluate the safety and efficacy of HIIT in terms of body shape, CRF, and cardiovascular disease biomarkers for children and adolescents of all ages (including health, obesity and disease) by synthesizing 47 eligible randomized controlled trials. 47 studies were assessed using strict inclusion and exclusion criteria (age, exercise intensity, exercise frequency and exercise time). Available evidence shows that HIIT can significantly improve most indicators of health in children and adolescents, including CRF indices (VO₂max, SBP, DBP and HR_max) and cardiovascular disease biomarkers (TG and HDL-C). In addition, there is insufficient evidence that HIIT improves body shape indicators (BMI, BF% and WC) and some cardiometabolic measures (TG, LDL-C) in children and adolescents.

Statistical heterogeneity consisted in most outcome measures owing to multiple factors. First, our meta-analysis included children and adolescents across a wide age range. Since this stage is prepubertal and adolescence, developmental speed is related to age. Previous studies have shown that prepubertal children may obtain greater benefits in HIIT [2], and that different developmental stages may affect the assessment of HIIT in body shape indicators. Second, the participants included in the study were healthy, overweight, obese, and had partial diseases,

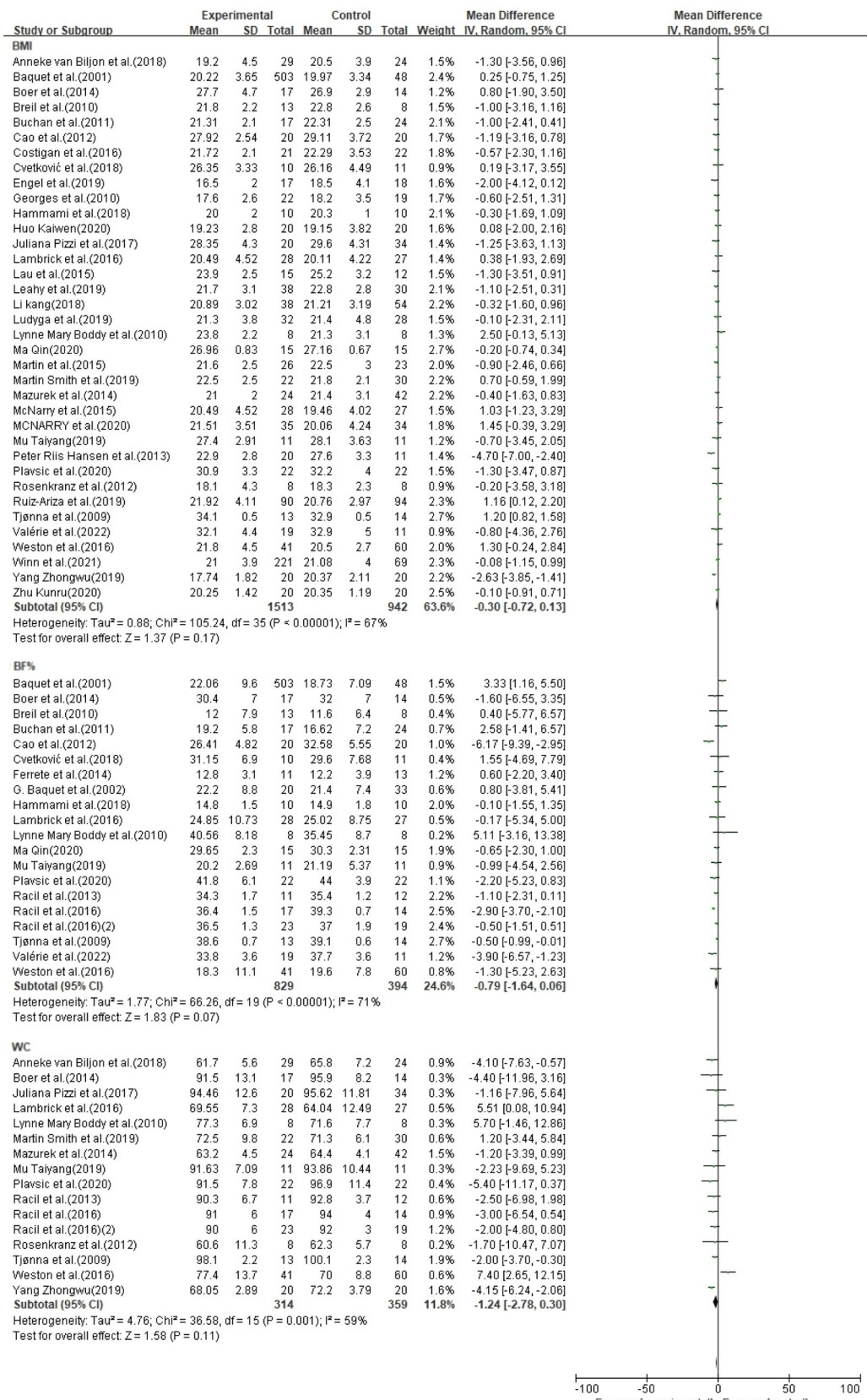

**Fig 3. Forest plot of meta-analysis on the effect of body morphology indicators.**

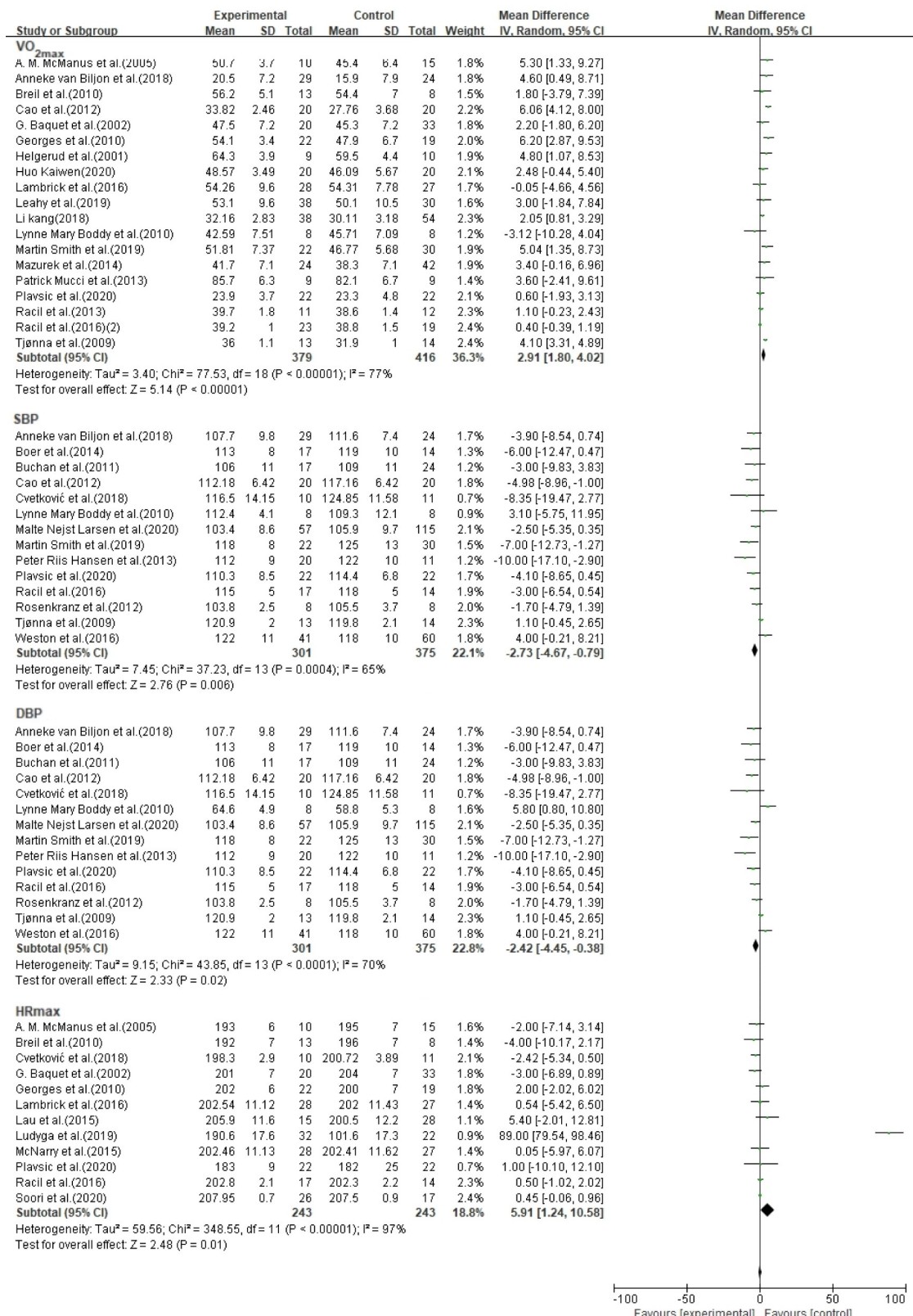

**Fig 4. Forest plot of meta-analysis on the effect of CRF indicators.**

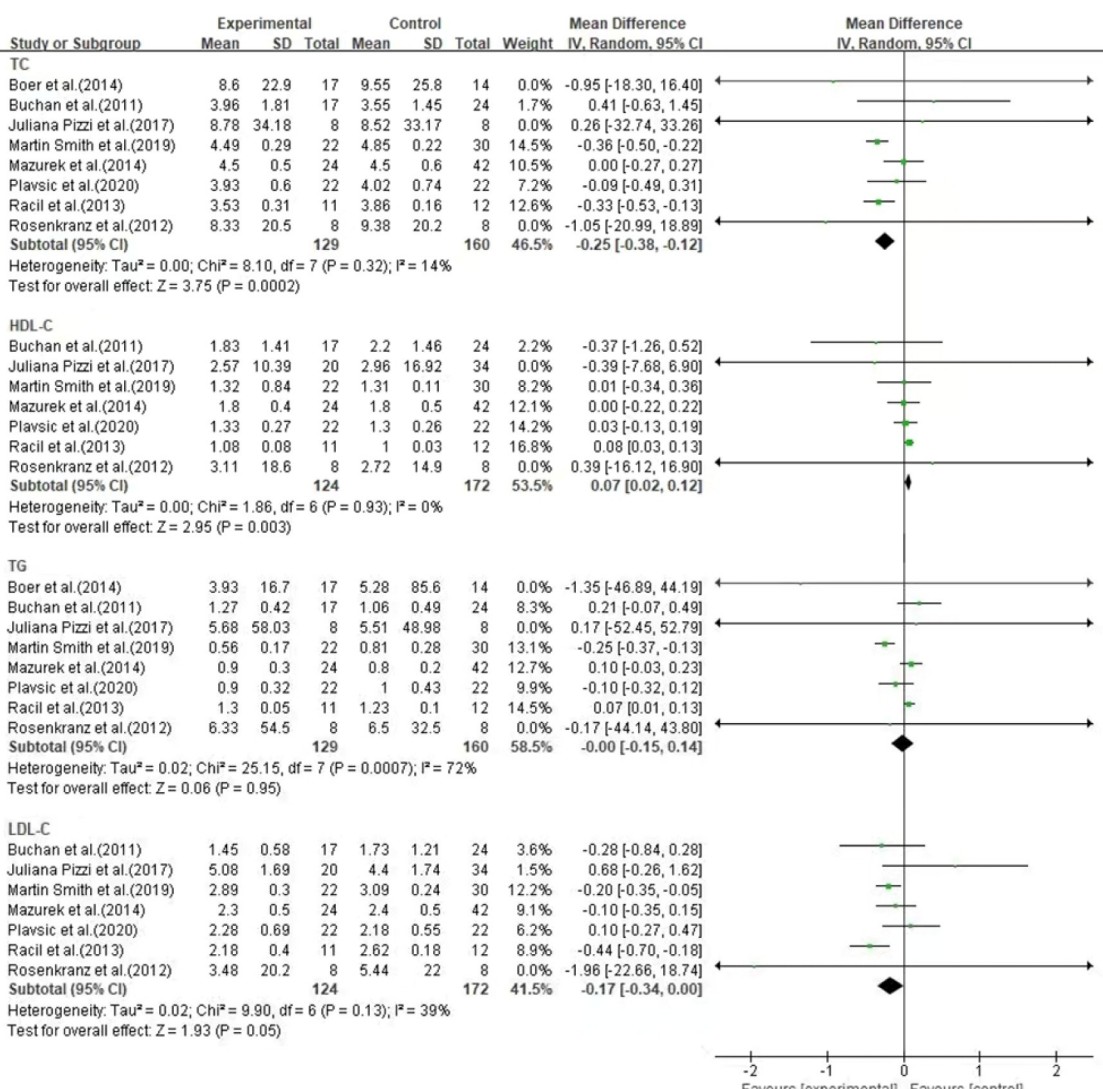

**Fig 5. Forest plot of meta-analysis on the effect of cardiovascular disease biomarkers.**

together with children and adolescents with sports training experience, which led to differences in participants at baseline level and affected the assessment of HIIT outcomes. Third, the study intervention time ranges from 1.57 weeks to 40 weeks, and it is generally believed that a longer exercise intervention time is more likely to obtain greater benefits. Finally, the exercise frequency of participants is an essential part of exercise prescription or exercise programs, and the exercise frequency of the included studies varied from 2 to 5 times/week, which may affect the evaluation of HIIT outcomes.

In addition, this study also determined the HIIT dose-response relationship: with interval running as the primary form of exercise, exercise intensity was $\geq$ 80% $VO_{2max}$ / $\geq$ 100% MAS / $\geq$ 80% $HR_{max}$. For healthy children and adolescents aged 5–15 years, the health benefit was the greatest when the intervention time was $\leq$ 10 weeks and the exercise frequency was 2–5 times/week, while children and adolescents aged $\geq$ 15 years could not recommend exercise doses because of the limited number of included studies.

## Effect of HIIT on body shape of children and adolescents

Previous studies have shown that HIIT can effectively improve body shape indicators in children and adolescents. Currently, there is insufficient evidence that HIIT improves body shape indicators, including BMI, BF% and WC in children and adolescents. The findings are consistent with previous studies [2, 3], but there are diametrically opposite conclusions [4]. Although this study showed that HIIT had no significant effect on the body shape indicators of children and adolescents, it may have a greater impact on the reliability of the results due to the large range of subjects included in the study. Further analysis revealed that HIIT had a significant effect on BF % (MD = -1.59, 95% CI [-2.59, -0.58], p = 0.002) (Fig 6) and WC (MD = -2.06, 95% CI [-3.26, -0.86]) (p < 0.001) in obese children and adolescents (Fig 7). The effect of the study results had no significant effect on BMI (MD = -0.91, 95% CI [-1.91, 0.09], p = 0.08). Metabolic disorders caused by overweight/obesity are the pathological basis of various metabolic diseases such as $T_2DM$ and CVD. Adverse metabolic phenotypes are highly associated with obesity in both children and adolescents. Unfortunately, more than 50% of children and adolescents develop obesity into adulthood, and this proportion increases with age [56]. In addition, children and adolescents with increased BMI have an increased risk of $T_2DM$, stroke, coronary heart disease and cancer. In addition, prospective cohort studies have shown that BF % and WC, two body shape indicators, dropped to the normal range, significantly reducing the prevalence of obesity-related diseases [57]. Therefore, a healthy body shape in children and adolescents is crucial.

At present, the key to the childhood obesity epidemic are inflammation and metabolic disorders. Anti-inflammatory and antioxidant interventions help to regulate inflammation and

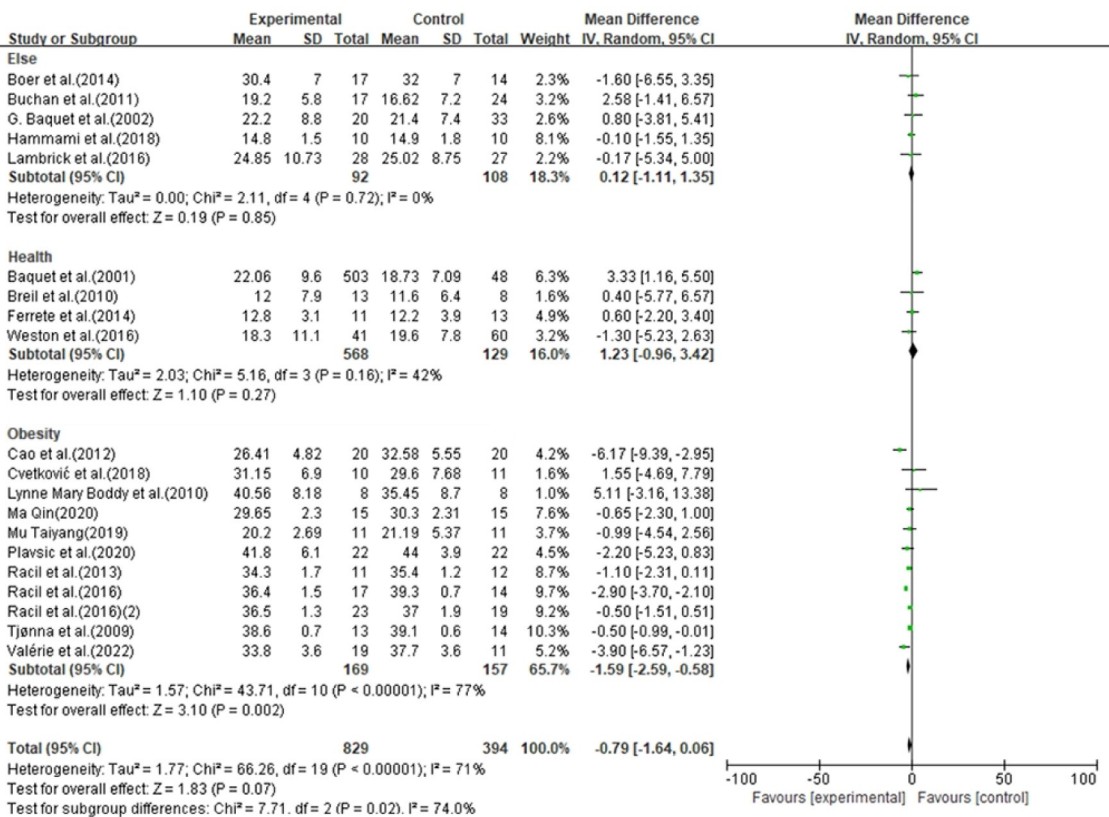

**Fig 6. Subgroup analysis of participants in children and adolescents with BF% in HIIT and control group.**

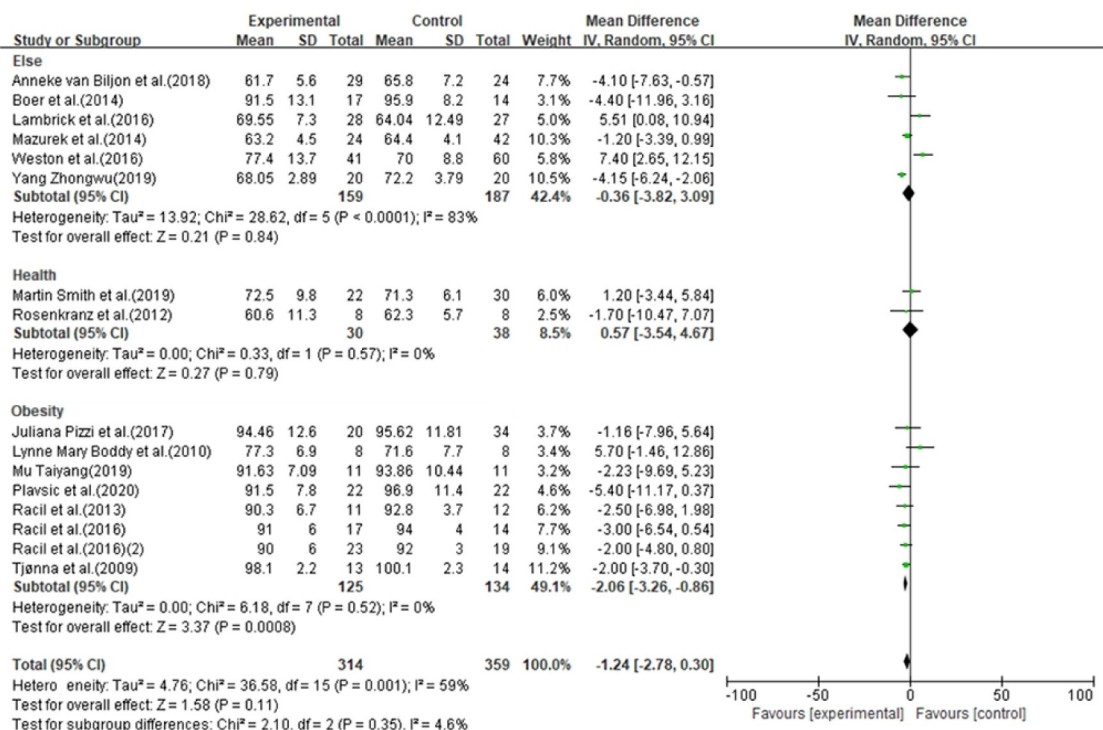

**Fig 7. Subgroup analysis of participants in children and adolescents with WC in HIIT and control group.**

metabolic disorders caused by obesity. Significant progress has been made in the prevention and treatment of obesity; however, obesity-related diseases still cannot be effectively controlled. It is worth noting that the majority of childhood and adolescent obesity cases are caused by insufficient physical activity. The prevalence of obesity in the physically inactive population has increased and there is still a potential risk in physically inactive non-obese individuals. However, the good news is that we found that regular exercise can independently reduce the risk of obesity, $T_2DM$, CVD, and cancer due to metabolic disorders, playing an important role in related diseases [58]. Particularly early exercise interventions are more effective for children and adolescents. Recent studies have shown that running-based HIIT can improve the physical health of obese adolescents, and a larger survey of participants from different schools is required to validate this conclusion [5].

Furthermore, we found a subset of athletes among healthy children and adolescents [25, 39, 47]. The research subjects were other children and adolescents with partial intellectual disability [38] and non-uniform subjects (the included participants were overweight, obese and healthy children and adolescents, and the data could not be classified and extracted) [30, 32]. There may be certain data risks due to inconsistencies in the population classification of study subjects.

The sensitivity analysis of this meta-analysis resulted in high stability, and we can conclude that although HIIT does not improve body shape indicators in children and adolescents, its recommendation for obese children and adolescents should be retained.

## Effect of HIIT on CRF in children and adolescents

CRF has become one of the most extensive components of physical health research, owing to its strong correlation with health outcomes. Strong epidemiological evidence suggests that

CRF is inversely associated with a high incidence of CVD, all-cause mortality and cancer in healthy populations. CRF can be used as a predictor of cancer mortality, and higher CRF levels can independently reduce cancer mortality in women and men [59]. This study confirmed that HIIT is an effective method for improving CRF indicators in children and adolescents, and the findings are consistent with those of most meta-analyses. Because of the differences in whether research subjects are obese or not, there is still controversy about HIIT replacing MICT [4]; however, it has gradually become widely accepted because HIIT is more cost-effective. Although the relationship between physical activity and the incidence and mortality of CVD, $T_2DM$ and cancer, etc. and its preventive effects have been demonstrated, children and adolescents still face the plight of insufficient physical activity. Extensive research has been conducted in the field of sports medicine, and many research results have been obtained.

Our meta-analysis results showed that HIIT can effectively improve the CRF index, but there was some statistical heterogeneity. We conducted a subgroup analysis of CRF index heterogeneity and found that participants and intervention time were the source of $HR_{max}$ heterogeneity, whereas exercise frequency was a source of heterogeneity in DBP and HRmax. However, age, participants, intervention time and exercise frequency were not heterogeneous sources of $VO_{2max}$ and SBP.

Due to the impaired oxidative metabolism of skeletal muscle in obese children and adolescents, HRmax exhibits considerable limitations in physical activity. Considering the effect of body weight on HRmax, the overall baseline level of HRmax was lower in obese children and adolescents and the magnitude of change was more sensitive to the HIIT intervention, compared to healthy children and adolescents where the change in HRmax levels tended to be stable. Furthermore, the proportion of obese adolescents in this subgroup analysis was more [11, 31, 33, 44]. Children, for example, are more resilient and have better tolerance in HIIT and seem to be more fatigue resistant than adolescents. More importantly, HIIT is more closely related to children's exercise habits, combined with children's exercise patterns (games) [30] and reward mechanisms [48, 50, 51] in the included studies, which make pre-adolescent children more autonomous and motivated to participate.

It is worth noting that high-intensity training triggers autonomic nervous disturbances and high exercise frequency triggers fatigue accumulation. The subgroup analysis included 6 studies [13, 31, 36, 46, 47, 50] with exercise frequency $\geq$ 3 times/week. Since children's tolerance, autonomy, and motivation to participate are more advantageous than those of adolescents. Future research should focus on the dose-response relationship between exercise frequency and CRF and the interest and reward mechanism of HIIT in prepubertal children during the specific implementation process. Simultaneously, improve the design of HIIT for adolescents to enhance their initiative and motivation to participate. Low frequency HIIT may be a viable and effective strategy for prescribing an initial exercise programme.

According to the available evidence, we can conclude that intervention time $\leq$ 10 weeks, frequency of exercise $\geq$ 3 times/week, and else children and adolescents aged 5–15 years are more sensitive to improving DBP after HIIT intervention. Intervention time > 10 weeks, frequency of exercise < 3 times/week, and healthy children and adolescents aged 5–15 years were more sensitive to improving $HR_{max}$ after HIIT intervention. Although partial results failed to explain the source of heterogeneity, DBP and $HR_{max}$, which are considered to be the most important outcome indicators for evaluating CRF, explained the heterogeneity well. It has been shown that HRmax has the potential to predict $VO_{2max}$, coupled with the fact that the methods used to measure SBP and DBP are the same, although failing to explain some of the heterogeneity that exists between $VO_{2max}$ and SBP. In summary, HIIT is recommended to improve CRF indicators in both children and adolescents.

## Effect of HIIT on metabolic risk factors of CVD in children and adolescents

At present, insufficient physical activity, unreasonable dietary structure, obesity, metabolic syndrome and other metabolic risk factors for cardiovascular disease are gradually superimposed, resulting in a sharp increase in the risk of CVD and $T_2DM$. Metabolic risk factors of cardiovascular disease in children and adolescents during this period have a significant influence on the onset of adulthood [56]. The good news is that cardiovascular risk factors are largely preventable, especially the effectiveness of exercise in improving metabolic risk markers for cardiovascular diseases is supported by substantial evidence.

According to our research, a meta-analysis of HIIT interventions in children and adolescents showed that HIIT was effective in improving TC and HDL-C, which are metabolic risk factors of cardiovascular disease in children and adolescents; however, the effect on TG and LDL-C appeared to be insignificant. The results of this meta-analysis are similar to the previous meta-analysis results [2], but there are some differences in the analysis of HIIT evaluation on children and adolescents in another study [3], which may be caused by HIIT's limited improvement of healthy children and adolescent blood lipids [37]. Most of the crowds we have included are healthy children and adolescents, and their incorporations are mostly obese children and adolescents.

Our convergence analysis results showed that HIIT can effectively improve TC and HDL-C levels without significant heterogeneity. Racil [40] and Boer et al. [38] reported that HIIT reduced TC in children and adolescents with a clinically significant (P < 0.05) and low risk of bias assessment, which are encouraging findings. Notably, Racil et al. [40] evaluated obese girls and Boer et al. [38] evaluated children and adolescents with disabilities. Racil [40] and Juliana et al. [28] reported that HIIT reduced HDL-C levels in children and adolescents with clinical significance (p < 0.05). Racil et al. [40] and Plavsic et al. [11] were evaluated in obese girls. The limitations of study methods and subjects for TG and HDL-C in the included studies may confuse the results, so caution should be exercised in interpreting these results. Although HIIT does not improve TG and LDL-C levels, the importance of TG level as an independent risk factor for CVD cannot be ignored. The effect of HIIT on LDL-C level (MD = -0.17, 95% CI [-0.34, 0.00], p = 0.05)statistically significant. After excluding the included studies individually, it was found that the study by Juliana et al. [28] had a high risk. After exclusion, the effect of HIIT on LDL-C level was statistically significant (p < 0.001), and there was no significant heterogeneity ($I^2$ = 24%, p = 0.25).

The results of the sensitivity analysis in this meta-analysis were highly stable and showed no significant heterogeneity. We can conclude that HIIT is effective in improving TC and HDL-C levels in children and adolescents, with little effect on TG and LDL-C levels. Considering the small sample size included in this meta-analysis, future research requires further expansion.

## Assessment of diet and leisure-time physical activity

35 studies [8–10, 12, 14–30, 34, 36, 38, 42–48, 50–53] did not describe dietary assessment in detail. 12 studies [7, 11, 14, 31–33, 35, 37, 39–41, 49] reported no changes in dietary habits. 35 studies [7–10, 12, 13, 15–23, 25–30, 34, 36, 38, 40, 43–48, 50–53] did not describe leisure-time physical activity in detail, while 12 studies [11, 14, 24, 31–33, 35, 37, 39, 41, 42, 49] conducted detailed assessments of leisure-time physical activity. Although 74.47% of the studies did not describe diet in detail, it can be understood as maintaining the original eating habits and physical activities by reading the full text. Furthermore, although 25.53% of the participants were assessed for dietary habits and leisure-time physical activity, the impact of dietary and leisure-time physical activity assessments on outcomes was not elucidated. It is worth noting that

maintaining the original dietary habits and physical activity helps rule out the influence of diet on HIIT outcomes; simultaneously, diet and leisure physical activity may be important limiting factors and sources of bias.

## Adverse events and compliance

In this meta-analysis, although only 21 studies [7, 11, 13, 17, 21–24, 32, 33, 35, 38–40, 43, 45–47, 49, 50, 53] (42.55%) reported withdrawal events due to family and subjective will (193 people dropped out, accounting for 6.445 of the total study, with a dependency of 93.56%). In addition, there were sicked children and adolescents in the included studies [8, 10, 13, 38], but no adverse events occurred, and safety and dependence were good. A previous meta-analysis of HIIT reported the occurrence of adverse events such as leg discomfort, joint sprains, asthma, myocardial infarction, etc [60]. The occurrence of adverse events was attributed to the study subjects belonging to high-risk groups for adverse events, such as patients with coronary heart disease and hypertension, but there seemed to be no difference in adverse events between the HIIT and control group. Medical supervision was described in 55.32% of the articles included in this meta-analysis. Therefore, these factors may cause biases, but the results may be skewed towards more positive effects.

## Advantages and limitations

Advantages of this study are as follows: (i) Retrieval was not limited by publication date. (ii) Research participants were not limited to specific children and adolescents, but included all children and adolescents, regardless of health, disease, etc. (iii) Subgroup analysis was carried out to explain the heterogeneity of research results, especially the analysis of exercise dose variables such as age (prepubertal and adolescence), research subjects, intervention time, and exercise frequency, which have often been ignored in previous studies. (iv) blank control was excluded from this study, and unbalanced results caused by non-blank control were avoided.

   Limitations of this study: (i) Although this review strictly implemented the retrieval strategy, due to limited conditions, only the literature published in Chinese and English was retrieved, and there may still be some publication bias due to the lack of a small number of published studies. (ii) This study only included information on children and adolescents in school, but lacked data about children and adolescents outside of school, which may have influenced the conclusion. (iii) quality of the included studies may be another factor; 12.77% of the studies did not use the randomized control model, and 17.02% of the studies reported the randomized control model but did not describe the randomized process. (iv) Although guidelines for HIIT have been established, the details of some guidelines still require refinement. (v) The biggest limiting factor may be that the age span of the individuals included in the study was large, important influencing factors such as exercise intensity, frequency, and time were not completely consistent, and the heterogeneity was considerable. (vi) None of the results from the subgroup analysis using age as a covariate were significant factors for heterogeneity due to study shortcomings, and some of the programmatic details suggested in the recommended guidelines for the explored HIIT program remain unclear.

   Although subgroup analyses were performed, some results still could not clarify the source of the heterogeneity. For example, sexual differences in developmental rates during childhood and adolescence (due to the mixed gender or too small sample size for subgroup analysis), different methods of outcome measures, and diet and leisure-time physical activity may all be sources of heterogeneity. In addition, the included studies did not disaggregate by sex, and the impact of gender on children and adolescents was still unclear; therefore, gender differences should be fully considered in future research.

## Conclusion

HIIT is safe, effective and less time-consuming for both child and adolescent health. Owing to its potential to improve body shape, CRF, and cardiovascular disease risk markers, it should be incorporated into the daily management of physical activity in children and adolescents. More importantly, the effect of HIIT has a higher consistency in gender, population, and age (pre-adolescence and adolescence); therefore, it has a higher generality in improving physical health. Although there were dropouts and loss to follow-up during this process, no adverse events caused by HIIT occurred. These findings highlight the potential role of HIIT in improving the health of both children and adolescents. Considering the lack of more detailed standards for HIIT interventions in the included studies, it is worth studying specific HIIT interventions (optimal exercise interval time and interval intensity) in different ages, sexes, and participants to make HIIT more effective and scientific.

In conclusion, strengthening medical supervision and adequate warm-up before exercise are more feasible for the promotion of HIIT in children and adolescents.

## Supporting information

**S1 Table. Search strategy.**
(DOCX)

**S2 Table. Excluded studies and reason for exclusion.**
(DOCX)

**S3 Table. Detailed table of basic characteristics of included studies.**
(DOCX)

**S4 Table. Measurement method(s) and distribution of outcome indicators.**
(DOCX)

**S1 Fig. Sensitivity analysis of BMI.**
(DOCX)

**S2 Fig. Sensitivity analysis of BF%.**
(DOCX)

**S3 Fig. Sensitivity analysis of WC.**
(DOCX)

**S4 Fig. Sensitivity analysis of TC.**
(DOCX)

**S5 Fig. Sensitivity analysis of TG.**
(DOCX)

**S6 Fig. Sensitivity analysis of HDL-C.**
(DOCX)

**S7 Fig. Sensitivity analysis of LDL-C.**
(DOCX)

**S8 Fig. Sensitivity analysis of $VO_{2max}$.**
(DOCX)

**S9 Fig. Sensitivity analysis of SBP.**
(DOCX)

**S10 Fig. Sensitivity analysis of DBP.**
(DOCX)

**S11 Fig. Sensitivity analysis of HR$_{max}$.**
(DOCX)

**S12 Fig. Subgroup analysis of age in children and adolescents with VO$_{2max}$ in HIIT and control group.**
(DOCX)

**S13 Fig. Subgroup analysis of participants in children and adolescents with VO$_{2max}$ in HIIT and control group.**
(DOCX)

**S14 Fig. Subgroup analysis of intervention time in children and adolescents with VO$_{2max}$ in HIIT and control group.**
(DOCX)

**S15 Fig. Subgroup analysis of exercise frequency in children and adolescents with VO$_{2max}$ in HIIT and control group.**
(DOCX)

**S16 Fig. Subgroup analysis of age in children and adolescents with SBP in HIIT and control group.**
(DOCX)

**S17 Fig. Subgroup analysis of participants in children and adolescents with SBP in HIIT and control group.**
(DOCX)

**S18 Fig. Subgroup analysis of intervention time in children and adolescents with SBP in HIIT and control group.**
(DOCX)

**S19 Fig. Subgroup analysis of exercise frequency in children and adolescents with SBP in HIIT and control group.**
(DOCX)

**S20 Fig. Subgroup analysis of age in children and adolescents with DBP in HIIT and control group.**
(DOCX)

**S21 Fig. Subgroup analysis of participants in children and adolescents with DBP in HIIT and control group.**
(DOCX)

**S22 Fig. Subgroup analysis of intervention time in children and adolescents with DBP in HIIT and control group.**
(DOCX)

**S23 Fig. Subgroup analysis of exercise frequency in children and adolescents with DBP in HIIT and control group.**
(DOCX)

**S24 Fig. Subgroup analysis of age in children and adolescents with HR$_{max}$ in HIIT and control group.**
(DOCX)

**S25 Fig. Subgroup analysis of participants in children and adolescents with HR$_{max}$ in HIIT and control group.**
(DOCX)

**S26 Fig. Subgroup analysis of intervention time in children and adolescents with HR$_{max}$ in HIIT and control group.**
(DOCX)

**S27 Fig. Subgroup analysis of exercise frequency in children and adolescents with HR$_{max}$ in HIIT and control group.**
(DOCX)

**S1 Checklist. The PRISMA checklist of current meta-analysis.**
(DOCX)

## Author Contributions

**Data curation:** Shuangling Zou, Jia Ma, Chenmin Xiang.

**Funding acquisition:** Jie Men.

**Methodology:** Jie Men, Shufeng Li.

**Project administration:** Jie Men.

**Software:** Shuangling Zou, Chenmin Xiang, Junli Wang.

**Supervision:** Jie Men, Shufeng Li.

**Writing – original draft:** Jie Men, Shuangling Zou, Jia Ma.

**Writing – review & editing:** Shufeng Li.

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
