## [Decision Letter · Decision Letter 0]

21 Oct 2022

PONE-D-22-19095High-Intensity Interval Training Improves physical morphology, Cardiopulmonary Fitness and Metabolic Risk Indicators of Cardiovascular Disease in Children and Adolescents: A Systematic Review and Meta-Analysis.PLOS ONE

Dear Dr. Jie Men,

Thank you for submitting your manuscript to PLOS ONE. After careful consideration, we feel that it has merit but does not fully meet PLOS ONE’s publication criteria as it currently stands. Therefore, we invite you to submit a revised version of the manuscript that addresses the points raised during the review process.

ACADEMIC EDITOR:Dear Author, Based on the comments provided by the reviewers, this manuscript still requires some major revision. Please attend to all the reviewers' comments and make the necessary changes. The decision of this manuscript is justified based on PLOS ONE’s publication criteria and not on its novelty or perceived impact.

We look forward to receiving your revised manuscript.

Kind regards,

Zulkarnain Jaafar

Academic Editor

PLOS ONE

Journal Requirements:

When submitting your revision, we need you to address these additional requirements. 1. Please ensure that your manuscript meets PLOS ONE's style requirements, including those for file naming. The PLOS ONE style templates can be found at https://journals.plos.org/plosone/s/file?id=wjVg/PLOSOne_formatting_sample_main_body.pdf and https://journals.plos.org/plosone/s/file?id=ba62/PLOSOne_formatting_sample_title_authors_affiliations.pdf 2. You stated that the "retrieval limit was from the establishment of the database to January 1, 2022" (lines 76-77) and that "3,868 studies were retrieved in December 2021" (line 120). Please clarify in your Methods section when the literature search was performed by providing the exact date. 3. Thank you for stating the following in the Acknowledgments Section of your manuscript: "The completion of the thesis is very grateful to the co-funding of the Teaching Reform and Innovation Project of Shanxi Provincial Department of Education (J2021967) and the Teaching Reform and Innovation Project of Fenyang College of Shanxi Medical University (FJ202013). Many thanks to the authors for their hard work"We note that you have provided additional information within the Acknowledgements Section that is not currently declared in your Funding Statement. Please note that funding information should not appear in the Acknowledgments section or other areas of your manuscript. We will only publish funding information present in the Funding Statement section of the online submission form. Please remove any funding-related text from the manuscript and let us know how you would like to update your Funding Statement. Currently, your Funding Statement reads as follows: "Yes. The completion of the thesis is very grateful to the co-funding of the Teaching Reform and Innovation Project of Shanxi Provincial Department of Education (J2021967) and the Teaching Reform and Innovation Project of Fenyang College of Shanxi Medical University (FJ202013)." Please include your amended statements within your cover letter; we will change the online submission form on your behalf.  4. Thank you for stating the following financial disclosure: ""Yes. The completion of the thesis is very grateful to the co-funding of the Teaching Reform and Innovation Project of Shanxi Provincial Department of Education (J2021967) and the Teaching Reform and Innovation Project of Fenyang College of Shanxi Medical University (FJ202013)."" 
Please state what role the funders took in the study.  If the funders had no role, please state: "The funders had no role in study design, data collection and analysis, decision to publish, or preparation of the manuscript." If this statement is not correct you must amend it as needed. Please include this amended Role of Funder statement in your cover letter; we will change the online submission form on your behalf.  5. We note that you have stated that you will provide repository information for your data at acceptance. Should your manuscript be accepted for publication, we will hold it until you provide the relevant accession numbers or DOIs necessary to access your data. If you wish to make changes to your Data Availability statement, please describe these changes in your cover letter and we will update your Data Availability statement to reflect the information you provide. 6. Could you please clarify the Table 2 Basic features of the included studies and Table 2 Three-line table of Egger's Publishing Bias? 7. Please include a copy of Table 3 which you refer to in your text on page 7. 8. Please include captions for your Supporting Information files at the end of your manuscript, and update any in-text citations to match accordingly. Please see our Supporting Information guidelines for more information: http://journals.plos.org/plosone/s/supporting-information

Reviewers' comments:

Reviewer's Responses to Questions

**Comments to the Author**

1. Is the manuscript technically sound, and do the data support the conclusions?

Reviewer #1: Partly

Reviewer #2: Partly

2. Has the statistical analysis been performed appropriately and rigorously? 

Reviewer #1: Yes

Reviewer #2: Yes

3. Have the authors made all data underlying the findings in their manuscript fully available?

Reviewer #1: Yes

Reviewer #2: Yes

4. Is the manuscript presented in an intelligible fashion and written in standard English?

Reviewer #1: No

Reviewer #2: No

5. Review Comments to the Author

Reviewer #1: The title is not correct, erroneously suggesting a beneficial effect of HIIT on all cardiometabolic indicators.

Numerous editorial errors indicating careless preparation of the manuscript. Conclusions not entirely correct.

In the opinion of the reviewer, the age range of the analysed group is too wide for a single metanalysis. This type of review should be carried out for at least three age groups, e.g., preschool and early school age children, high school students and young adults.

Reviewer #2: 1. Title should have to be in sentences case

2. Author and affiliations should have to be follow the journal guideline

3. Corresponding author missed

4. Abstract should be well oriented

5. Text indentations, revise the English grammar and capitalization

6. Systematic review title should have to be checked if others are running it through PROSPERO. Hence, this should be explained under methods part.

7. Discussion part - Th first paragraph of the last statements (line 214-218) should be cited.

8. Line 219 "This study is the first to systematically..." are you sure for this? (what about this study? DOI: 10.1101/2022.07.11.22277515)

9. Systematic review needs tremendous effort. The searching methods should be well oriented and include all the studies. Hence, following PLOS ONE guideline, this paper should have to be amended well to be published since it is a good finding.

6. PLOS authors have the option to publish the peer review history of their article (what does this mean?). If published, this will include your full peer review and any attached files.

Reviewer #1: No

Reviewer #2: No

---

## [Author Response · Author response to Decision Letter 0]

18 Dec 2022

Deare editor and reviewers，

Thank you for offering us an opportunity to improve the quality of our submission manuscript (PONE-D-22-19095). We appreciated very much the reviewers’ constructive and insightful comments. In this revision, we have addressed all these comments and suggestions. We hope that the revised manuscript has now met the publication standard of your journal. We highlighted all the revisions in red colour.

Firstly, the following changes have been made to meet the additional requirements of this journal:

1.We have corrected incorrect words and grammar to meet the rules of standard English writing wherever possible.

2. Support & Fund Information：Teaching Reform and Innovation Project for Higher Education Institutions of Shanxi Provincial Education Department (No.J2021967)、Teaching Reform Innovation Project of Fenyang College of Shanxi Medical University (No.FJ202013) and 2022 Research Project of Fenyang College of Shanxi Medical University (No.2022A01)。FJ202013 and 2022A01 provide platform resources in terms of data collection and analysis.J2021967 provided constructive advice on publication decisions and manuscript preparation.

3. Table 2 “Basic features of the included studies” and Table 2 “Three-line table of Egger’s Publishing Bias” in the original manuscript were incorrectly converted by the software when converting from docx format to pdf format, resulting in an error in the table names.

Response to reviewer

Is the manuscript technically sound, and do the data support the conclusions?

Reviewer #1: Partly

Reviewer #2: Partly

Response：Based on the reviewers' suggestions and comments, we have re-run the statistical analysis of the data. In the discussion section, we start with the results of the available evidence, re-analyse potential sources of heterogeneity and discuss them in detail after identifying their significant sources of heterogeneity and compare them with previous studies (e.g. intervention methods, findings), especially the parts with different or controversial findings. Given that there are still some limitations to this article, the specific exercise interventions we recommend are only an objective suggestion (see the discussion section for more details).

Reviewer #1: The title is not correct, erroneously suggesting a beneficial effect of HIIT on all cardiometabolic indicators.

Numerous editorial errors indicating careless preparation of the manuscript. Conclusions not entirely correct.

Response：The corrected title is “Effects of high-intensity interval training on physical morphology, cardiopulmonary fitness and metabolic risk indicators of cardiovascular disease in children and adolescents: a systematic review and meta-analysis.”

In the opinion of the reviewer, the age range of the analysed group is too wide for a single metanalysis. This type of review should be carried out for at least three age groups, e.g., preschool and early school age children, high school students and young adults.

Response：Our division into age groups was based on the actual situation of the included studies. If the studies were divided into at least three groups as suggested by the reviewers, there would be only four relevant studies in which the age stage of the study population belonged to young people, which is a smaller sample size than the other groups and insufficient to support the results of a robust subgroup analysis. Therefore the subgroups in this subgroup analysis were grouped according to 5- 15 years and ≥15 years. Although the age range is too wide, more robust results can be obtained.

Reviewer #2: Systematic review title should have to be checked if others are running it through PROSPERO. Hence, this should be explained under methods part.

Response：We were registered with PROSPERO under the registration number 10.37766/inplasy 2022.10.0092.

Line 219 "This study is the first to systematically..." are you sure for this? (what about this study? DOI: 10.1101/2022.07.11.22277515)

Response：This article (DOI: 10.1101/2022.07.11.22277515) is a preprint version of the manuscript that we published in medRxiv (https://medrxiv.org/cgi/content/short/2022.07.11.22277515v1). 

PLOS authors have the option to publish the peer review history of their article (what does this mean?). If published, this will include your full peer review and any attached files.

Do you want your identity to be public for this peer review? For information about this choice, including consent withdrawal, please see our Privacy Policy.

Reviewer #1: No

Reviewer #2: No　　

Response：Consent for my identity to be made public in peer review.

I hope you are satisfied with our changes. If you have any questions, please be sure to let me know, I will reply to you as soon as possible after receiving comments.

Declaration: The manuscript (in whole or part) has not been submitted or published in other journals at the same time. All authors have read and agreed to the content of the manuscript, and there is no potential conflict of interest.

Thank you for taking the time to read the manuscript again and consider my paper. 

Corresponding author, Jie Men, menjie2020@126.com，+86-18903588568.

---

## [Decision Letter · Decision Letter 1]

17 Jan 2023

PONE-D-22-19095R1Effects of high-intensity interval training on physical morphology, cardiopulmonary fitness and metabolic risk indicators of cardiovascular disease in children and adolescents: a systematic review and meta-analysis.PLOS ONE

Dear Dr. Jie Men,

Thank you for submitting your manuscript to PLOS ONE. After careful consideration, we feel that it has merit but does not fully meet PLOS ONE’s publication criteria as it currently stands. Therefore, we invite you to submit a revised version of the manuscript that addresses the points raised during the review process.

ACADEMIC EDITOR: Dear Author,This manuscript still requires some corrections to be made. Please make the necessary changes based on the comments provided by the reviewer/s. The decision of this manuscript is justified based on PLOS ONE’s publication criteria and not on its novelty or perceived impact.

We look forward to receiving your revised manuscript.

Kind regards,

Zulkarnain Jaafar

Academic Editor

PLOS ONE

Reviewers' comments:

Reviewer's Responses to Questions

**Comments to the Author**

1. If the authors have adequately addressed your comments raised in a previous round of review and you feel that this manuscript is now acceptable for publication, you may indicate that here to bypass the “Comments to the Author” section, enter your conflict of interest statement in the “Confidential to Editor” section, and submit your "Accept" recommendation.

Reviewer #1: (No Response)

Reviewer #2: All comments have been addressed

2. Is the manuscript technically sound, and do the data support the conclusions?

Reviewer #1: No

Reviewer #2: Yes

3. Has the statistical analysis been performed appropriately and rigorously? 

Reviewer #1: Yes

Reviewer #2: Yes

4. Have the authors made all data underlying the findings in their manuscript fully available?

Reviewer #1: Yes

Reviewer #2: Yes

5. Is the manuscript presented in an intelligible fashion and written in standard English?

Reviewer #1: Yes

Reviewer #2: Yes

6. Review Comments to the Author

Reviewer #1: I maintain my position negatively evaluating the analysis of data in the 5-15 age group defined as children. I believe that the analysis could possibly take into account young people aged 11-16 who are in adolescence.

In addition, the authors use the terms inconsistently: “cardiorespiratory fitness” (in text) versus “cardiopulmonary fitness” (in the title).

It is also a defect to define lipid profile indicators- TC, HDL-C, LDL-C (data evaluated by Authors) as “metabolic risk indicators of cardiovascular disease”. The term: “metabolic risk indicators of cardiovascular disease” used in the work erroneously suggests that such indicators were assessed as: blood glucose concentration, hypertension, concentration of serum hight-insensitivity C-reactive proteins, lipoprotein A etc.

The conclusions go absolutely too far.

Reviewer #2: The authors have included all the requested information. Hence, I am pleased to be published.

Registration for the PROSPERO has been done.

The methods are updated and discussion has been amended.

7. PLOS authors have the option to publish the peer review history of their article (what does this mean?). If published, this will include your full peer review and any attached files.

Reviewer #1: No

Reviewer #2: No

---

## [Author Response · Author response to Decision Letter 1]

7 Mar 2023

Dear editors and reviewers，

All the changes are highlighted in green to distinguish them from the first changes made in red.

After searching the PubMed MeSH database, we determined that all expressions related to “cardiorespiratory fitness”/ “cardiopulmonary fitness” in the text were corrected to “cardiorespiratory fitness”.

After reviewing the literature again, we found that it is not appropriate to simply use lipid metabolism indicators (TC, HDL-C, LDL-C), blood glucose concentration, hypertension, etc. as cardiometabolic indicators. Because the term "cardiometabolic index" is used incorrectly in the text, combining the extensive literature evidence: the most common cardiometabolic risk factors in the population include altered lipid profiles (TC, TG, HDL-C), altered blood pressure, and obesity. Furthermore, these factors tend to cluster together and often occur in combination in individuals [1,2,3], Therefore, we changed the term "cardiometabolic indicators" to "cardiometabolic risk factors," Rationalizes the use of lipid profiles, blood glucose concentrations, and hypertension as biomarkers of cardiovascular disease (i.e., outcome indicators).

After discussion, we found that our age subgroup analysis had some limitations(see the description of the limitations of the discussion). Previously, we also received this revision from you, tried what you said about dividing the covariate of age into at least three groups, but based on the basic characteristics of our included literature - there were only four relevant studies in which the age of the study subjects belonged to young adults - and the large age span of the subject groups themselves in most of the included studies (which could not be clearly grouped) made the sample size of some subgroups small enough to support the results of a robust subgroup analysis. Furthermore, differences in participant characteristics between pre-pubertal and pubertal studies with regard to baseline body mass or body composition and health status may confound the interpretation of inter-study differences and their attribution to maturity per se[3,4]. Therefore, few similar meta-analyses in the literature would use maturity (prepubertal and postpubertal) as a covariate for subgroup analysis, let alone for detailed grouping. The subgroup we provide as age, although broad in scope, also allows for a more comprehensive analysis of significant influences on heterogeneity with some credibility.

I hope you are satisfied with our changes. If you have any questions, please be sure to let me know, I will reply to you as soon as possible after receiving comments.

Reference:

1. Monserrat Solera-Martínez ÁHIM. High-Intensity Interval Training and Cardiometabolic Risk Factors in Children: A Meta-analysis. PEDIATRICS. 2021;148(4):e2021050810.

2. Liu J, Zhu L, Su Y. Comparative Effectiveness of High-Intensity Interval Training and Moderate-Intensity Continuous Training for Cardiometabolic Risk Factors and Cardiorespiratory Fitness in Childhood Obesity: A Meta-Analysis of Randomized Controlled Trials. FRONT PHYSIOL. 2020 2020-04-03;11.

3. Weston KL, Azevedo LB, Bock S, Weston M, George KP, Batterham AM. Effect of Novel, School-Based High-Intensity Interval Training (HIT) on Cardiometabolic Health in Adolescents: Project FFAB (Fun Fast Activity Blasts) - An Exploratory Controlled Before-And-After Trial. PLOS ONE. 2016 2016-08-03;11(8):e159116.

4. Eddolls WTB, McNarry MA, Stratton G, Winn CON, Mackintosh KA. High-Intensity Interval Training Interventions in Children and Adolescents: A Systematic Review. SPORTS MED. 2017;47(11):2363-74.

---

## [Decision Letter · Decision Letter 2]

27 Mar 2023

PONE-D-22-19095R2Effects of high-intensity interval training on physical morphology, cardiorespiratory fitness and metabolic risk factors of cardiovascular disease in children and adolescents: a systematic review and meta-analysis.PLOS ONE

Dear Dr. Jie Men,

Thank you for submitting your manuscript to PLOS ONE. After careful consideration, we feel that it has merit but does not fully meet PLOS ONE’s publication criteria as it currently stands. Therefore, we invite you to submit a revised version of the manuscript that addresses the points raised during the review process.

ACADEMIC EDITOR:Dear Author,1. Please make the necessary changes as suggested by the reviewer.2. Also please address this issue. Please justify in the manuscript why are you including these studies in your analysis in view of the participants age studies. Among the included are: Rosenkranz et al.[43] Mazurek et al.[37] Peter Riis Hansen et al.[41] Ferrete et al.[39] G. Baquet et al.[51] Lambrick et al.[30] . The decision of this manuscript is justified based on PLOS ONE’s publication criteria and not on its novelty or perceived impact.

We look forward to receiving your revised manuscript.

Kind regards,

Zulkarnain Jaafar

Academic Editor

PLOS ONE

Journal Requirements:

Reviewers' comments:

Reviewer's Responses to Questions

**Comments to the Author**

1. If the authors have adequately addressed your comments raised in a previous round of review and you feel that this manuscript is now acceptable for publication, you may indicate that here to bypass the “Comments to the Author” section, enter your conflict of interest statement in the “Confidential to Editor” section, and submit your "Accept" recommendation.

Reviewer #1: (No Response)

2. Is the manuscript technically sound, and do the data support the conclusions?

Reviewer #1: Yes

3. Has the statistical analysis been performed appropriately and rigorously? 

Reviewer #1: Yes

4. Have the authors made all data underlying the findings in their manuscript fully available?

Reviewer #1: Yes

5. Is the manuscript presented in an intelligible fashion and written in standard English?

Reviewer #1: Yes

6. Review Comments to the Author

Reviewer #1: There is a missing dot in table 2 in the age information: Lambrick et al.[30] 2016 Obesity: 93±0.8 Normal: 92±0.7

7. PLOS authors have the option to publish the peer review history of their article (what does this mean?). If published, this will include your full peer review and any attached files.

Reviewer #1: No

---

## [Author Response · Author response to Decision Letter 2]

20 Apr 2023

Dear Editors and Reviewers,

Thank you for providing us with the opportunity to improve the quality of your submissions (PONE-D-22-19095). We appreciate the constructive and insightful comments from the reviewers. In this revision, we have addressed all comments and suggestions. We hope that the revised manuscript has now met your publication's publishing standards. We have highlighted all revisions in yellow to distinguish the previous two revisions. 

First of all, we carefully read the reviewers' comments, and carefully read and analyzed the problematic literature you raised based on the review comments. 

According to the inclusion exclusion criteria for this paper, "Participants: Children and adolescents aged 5-19 years (normal weight, obesity, disease, etc.)." The above literature met the exclusion criteria. Six papers presented by the reviewers: Rosenkranz et al.[43] (experimental group: 8.8±0.6; control group: 9.8±4.1), Mazurek et al.[37] (experimental group: 19.5±0.6; control group: 19.5±0.6), Peter Riis Hansen et al.[41] (experimental group: 8-12 ; control group: 8-12), Ferrete et al.[39] (experimental group: 9.32±0.25; control group: 8.26±0.33), G. Baquet et al.[51] (experimental group:9.7± 0.9; control group: 10.1±0.4), Lambrick et al.[30] (experimental group:9.3± 0.8/9.2±0.7; Control Group: 9.4±0.8/9.2±0.8), all within the age range we included in the exclusion criteria. To assess the effects of H IIT more objectively, we did not limit participants. 

We evaluated the effects of HIIT in children and adolescents with the aim of scientifically and objectively reflecting the effects of HIIT on them. In addition, our age subgroup analysis of children and adolescents yielded robust results suggesting that the effects of HIIT on children and adolescents are widespread. 

In response to "There is a missing dot in table 2 in the age information: Lambrick et al.[30] 2016 Obesity: 93±0.8 Normal: 92±0.7", We have made corresponding changes in the original text and highlighted the changes. 

The above is our response to the reviewers' comments, if we have a deviation in the understanding of the reviewers' comments, or if you have any questions, we look forward to your further replies to deepen our understanding of the reviewers' comments and improve the quality of the paper. 

Declaration: The manuscript (in whole or part) has not been submitted or published in other journals at the same time. All authors have read and agreed to the content of the manuscript, and there is no potential conflict of interest.

Thank you for taking the time to read the manuscript again and consider my paper. 

Corresponding author, Jie Men, menjie2020@126.com，+86-18903588568.

---

## [Editor Report · Decision Letter 3]

24 Apr 2023

Effects of high-intensity interval training on physical morphology, cardiorespiratory fitness and metabolic risk factors of cardiovascular disease in children and adolescents: a systematic review and meta-analysis.

PONE-D-22-19095R3

Dear Dr. Jie Men,

We’re pleased to inform you that your manuscript has been judged scientifically suitable for publication and will be formally accepted for publication once it meets all outstanding technical requirements.

Kind regards,

Zulkarnain Jaafar

Academic Editor

PLOS ONE
---

## [Editor Report · Acceptance letter]

3 May 2023

PONE-D-22-19095R3 

Effects of high-intensity interval training on physical morphology, cardiorespiratory fitness and metabolic risk factors of cardiovascular disease in children and adolescents: a systematic review and meta-analysis. 

Dear Dr. Men:

I'm pleased to inform you that your manuscript has been deemed suitable for publication in PLOS ONE. Congratulations! Your manuscript is now with our production department. 

Kind regards, 

on behalf of

Dr. Zulkarnain Jaafar 

Academic Editor

PLOS ONE